# Neural Isometries:
# Taming Transformations for Equivariant ML

**Thomas W. Mitchel**
PlayStation
tommy.mitchel@sony.com

**Michael Taylor**
PlayStation
mike.taylor@sony.com

**Vincent Sitzmann**
MIT
sitzmann@mit.edu

## Abstract

Real-world geometry and 3D vision tasks are replete with challenging symmetries that defy tractable analytical expression. In this paper, we introduce Neural Isometries, an autoencoder framework which learns to map the observation space to a general-purpose latent space wherein encodings are related by *isometries* whenever their corresponding observations are geometrically related in world space. Specifically, we regularize the latent space such that maps between encodings preserve a learned inner product and commute with a learned functional operator, in the same manner as rigid-body transformations commute with the Laplacian. This approach forms an effective backbone for self-supervised representation learning, and we demonstrate that a simple off-the-shelf equivariant network operating in the pre-trained latent space can achieve results on par with meticulously-engineered, handcrafted networks designed to handle complex, nonlinear symmetries. Furthermore, isometric maps capture information about the respective transformations in world space, and we show that this allows us to regress camera poses directly from the coefficients of the maps between encodings of adjacent views of a scene.

## 1 Introduction

We constantly capture lossy observations of our world – images, for instance, are 2D projections of the 3D world. Observations captured consecutively in time are often related by transformations which are easily described in world space, but are intractable in the space of observations. For instance, video frames captured by a camera moving through a static scene are fully described by a combination of the 3D scene geometry and SE(3) camera poses. In contrast, the image space transformations between these frames can only be characterized by optical flow, a high-dimensional vector field that does not itself have any easily tractable low-dimensional representation.

Geometric deep learning seeks to build neural network architectures that are provably robust to transformations acting on their inputs, such as rotations [1–3], dilations [4, 5], and projective transformations [6, 7]. However, such approaches are only tractable for transformations that have group structure, and, even in those cases, still require meticulously hand-crafted and complex architectures. Yet, many real-world transformations of interest, for instance in vision and geometry processing, altogether lack identifiable group structure, such as the effect of camera motion in image space—see Fig. 1 to the right. Even when group-structured, they are often non-linear and non-compact, such as is the case of image

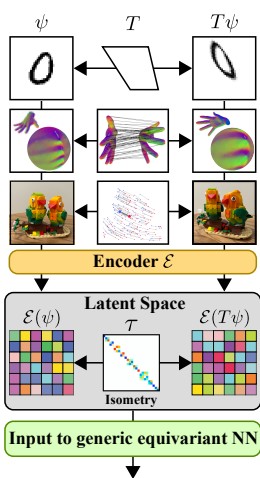

Figure 1: Neural Isometries find latent spaces where complex transformations become tractable.

38th Conference on Neural Information Processing Systems (NeurIPS 2024).

homographies and non-rigid shape deformations where existing approaches can be prohibitively expensive.

In this paper, we take a first step towards a class of models that learns to be equivariant to unknown and difficult geometric transformations in world space. We propose Neural Isometries (**NIso**), an autoencoder framework which learns to map the observation space to a latent space in which encodings are related by tractable, highly-structured linear maps whenever their corresponding observations are geometrically related in world space.

Specifically, observations are encoded into latents preserving their spatial dimensions. Images, for instance, can be encoded into latent functions defined over a lower resolution grid or the patch tokens of a ViT. For observations sharing some potentially unknown relationship in world space, we enforce their encodings be related by a functional map $\tau$ – a linear transformation on the space of latent functions. In particular, we require that $\tau$ is an *isometry* such that it preserves a learned inner product and commutes with a learned functional operator, in the sense that rigid body transformations commute with the Laplace operator in Euclidean space.

Neural Isometries exhibit unique properties that make them a promising step towards an architecture-agnostic regime for self-supervised equivariant representation learning. We experimentally validate two principle claims regarding the efficacy and applicability of our approach:

- Neural Isometries recover a general-purpose latent space in which challenging symmetries in the observation space can be reduced to compact, tractable maps in the latent space. We show that this can be exploited by simple isometry-equivariant networks to achieve results on par with leading hand-crafted equivariant networks in tasks with complex non-linear symmetries.

- The latent space constructed is geometrically informative in that it encodes information about the transformations in world space. We demonstrate that robust camera poses can be regressed directly from the isometric functional maps between encodings of adjacent views of a scene.

## 2   Related Work

**Geometric Deep Learning.**   Geometric deep learning is generally concerned with hand-crafting architectures that are equivariant to (*i.e.* that commute with) *known* transformations acting on the data [1–4, 6–17]. To this end, many successful architectures exploit group representations by using established mappings, such as the Fourier or spherical harmonic transforms, to map features onto domains where the group actions manifest equivalently as linear transformations [1–3, 7]. In most cases, the representations considered are finite-dimensional and *irreducible* which, loosely speaking, means that the group action in observation space can be expressed exactly by frequency-preserving block-diagonal matrices acting on the transform coefficients. While finite-dimensional irreducible representations (**IRs**) are attractive building blocks for equivariance due to their computationally exploitable structure, they often don't exist for non-compact groups, precluding generalizations to most non-linear symmetries, let alone those ill-modeled by groups. We instead avoid a heuristic choice of symmetry model and seek an approach that enables robustness to arbitrary transformations that may not even be group-structured.

**Functional Maps.**   Essential to our approach is the parameterization of transformations between observations in the latent space not as group representations, but instead as *functional maps*. Introduced in the seminal work of Ovsjanikov *et al.* [18], functional maps (**FMs**) provide a powerful medium for interpreting, manipulating, and constructing otherwise intractable mappings between 3D shapes via their realization as linear transformations on spaces of functions on meshes, forming the basis for state-of-the-art pipelines in shape matching and correspondence [19–23]. Beyond 3D shapes, FMs can be seen as a tool to parameterize transformations between observations viewed as functions, in the sense that images, for instance, are functions that map 2D pixel coordinates to RGB colors. Integral to their study and implementation is the Laplace-Beltrami operator (the generalization of the Laplace operator to function spaces on manifolds), and in particular the expression of FMs in its eigenbasis which can expose specific geometric properties of deformations. In particular, isometries (distance-preserving transformations) manifest as highly-structured matrices not unlike IRs, being orthogonal and commuting with the diagonal matrix of eigenvalues and are thus approximately block-diagonal. That said, FMs are recovered through regularized linear solves and as such lack the consistency inherent in the analytical expressions of IRs for compact groups. However, freed

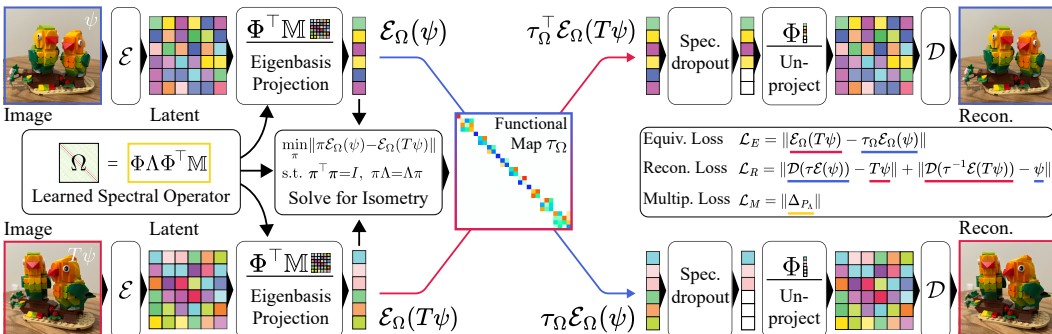

Figure 2: **Overview of Neural Isometries (NIso).** NIso learn a latent space where transformations of observations manifest as isometries, achieved by regularizing the functional maps $\tau$ between latents to commute with a learned operator $\Omega$, parameterized via its spectral decomposition into a mass matrix $\mathbb{M}$, eigenfunctions $\Phi$, and eigenvalues $\Lambda$ (sec. 4.1). Given two observations $\psi$ and $T\psi$ related by some unknown transformation $T$ (in this case, camera motion in a 3D scene), they are first encoded into latent *functions* $\mathcal{E}(\psi)$ and $\mathcal{E}(T\psi)$ and projected into the operator eigenbasis. An isometric functional map $\tau_\Omega$ is estimated between them, and used to map one to the other. Losses promote isometry-equivariance in the latent space, reconstruction of transformed latents, and distinct, low-multiplicity eigenvalues $\Lambda$, with the latter encouraging a diagonal as possible $\tau_\Omega$. An optional spectral dropout layer can be applied before the basis unprojection to encourage a physically meaningful ordering of the learned spectrum (sec. 4.2).

from nagging theoretical constraints, FMs have displayed remarkable representational capacity, well modeling a variety of highly-complex non-rigid deformations [24] including those without group structure, such as partial [25] and inter-genus correspondence [26]. Please see [27] for an outstanding introduction to functional maps.

**Discovering Latent Symmetries.** A number of recent approaches have proposed autoencoder frameworks wherein given symmetries in the base space manifest as simple operations in the latent space [28–35]. Perhaps most similar to our approach is recent work on a Neural Fourier Transform (NFT) that has sought to manifest group actions in observation space as IRs in the the latent space [36, 37]. These methods offer impressive theoretical guarantees under the conditions that the observed transformations are either known or are a group action, although these assumptions may not hold for real-world data with complex symmetries. In contrast, our method is wholly unsupervised, and assumes no knowledge of the transformations in observation space nor that they even form a group.

## 3   Method Overview

Neural Isometries are an architecture-agnostic autoencoder framework which learns to map pairs of observations related in world space to latents which are related by an approximately isometric FM $\tau$ – see Fig. 2. We first formulate encoding and decoding, and define a FM in the latent space. Next, we show how a functional operator $\Omega$ and mass matrix $\mathbb{M}$ can be learned in the latent space to regularize $\tau$ by requiring it to be an isometry. Specifically, for observations sharing some potentially unknown relationship in world space, we enforce there exist a FM $\tau$ between their encodings satisfying two key properties: 1) That $\tau$ preserves the functional inner product in the latent space determined by $\mathbb{M}$; and 2). $\tau$ *commutes* with the functional operator $\Omega$. Subsequently, we show that such maps can be recovered analytically through a differentiable, closed-form least squares solve in the operator eigenbasis. Last, we formulate NIso as an optimization problem incorporating both the strictness of the isometric correspondence between latents and the eigenvalue multiplicity of the operator, the latter of which controls the structure of the maps.

In experiments, we demonstrate that NIso are capable of both discovering approximations of known operators and constructing latent spaces where complex, non-linear symmetries in the observation space manifest equivalently as isometries. We show how the latter property can be exploited by demonstrating that a simple vector neuron MLP [38] acting in our pre-trained latent space can achieve results on par with state-of-the-art handcrafted equivariant networks operating in observation space. Subsequently, we consider the task of pose estimation, and demonstrate that robust SE(3) camera

poses can be extracted from latent transformations, serving as evidence that NIso encourages models to encode information about transformations in world space.

# 4 Neural Isometries

We consider an observation space $O \subset L^2(M, \mathbb{R}^n)$ consisting of functions defined over some domain $M$ (*e.g.* with $M$ the plane and $n = 3$ for RGB images). Here, elements of $O$ are in fact captures from some world space $W$ with $\sigma : W \to O$ representing the mechanism from which $O$ is formed from $W$. Furthermore, we assume there is a potentially unknown collection of phenomena $\{T\}$ acting on the world space that relates observations. That is, for some $w \in W$, $\psi = \sigma(w) \in O$, and denoting $T\psi \equiv \sigma(Tw)$, we assume that $T\psi$ is also in $O$ and that we are able to associate it with $\psi$.

Additionally, we consider an autoencoder consisting of an encoder and decoder

$$\mathcal{E} : L^2(M, \mathbb{R}^n) \to L^2(N, \mathbb{R}^d) \qquad \text{and} \qquad \mathcal{D} : L^2(N, \mathbb{R}^d) \to L^2(M, \mathbb{R}^n) \tag{1}$$

mapping between observation space and a space of latent functions over some domain $N$. In practice, we operate over discretizations of $M$ and $N$, with $\psi \in O$ and $\mathcal{E}(\psi) \in L^2(N, \mathbb{R}^d)$ represented as tensors $\psi \in \mathbb{R}^{|M| \times n}$ and $\mathcal{E}(\psi) \in \mathbb{R}^{|N| \times d}$. For example, if $M$ consists of the pixel indices of an image, then $N$ could be grid or token indices if $\mathcal{E}$ is a ConvNet or ViT, respectively.

**Goal: Equivariance of Latent Functions.** Our aim is to train the autoencoder such that for any $T$ acting in the world space and corresponding observations $\psi, T\psi \in O$, there exists a linear map $\tau : L^2(N, \mathbb{R}^d) \to L^2(N, \mathbb{R}^d)$ such that

$$\mathcal{E}(T\psi) \approx \tau \mathcal{E}(\psi). \tag{2}$$

In other words, we desire our latent space to be *equivariant* under world space transformations. As our problem is discrete, $\tau$ is a *functional map* – an $|N| \times |N|$ matrix representation of maps on $L^2(N, \mathbb{R}^d)$. In the case of latents with $|N| = H \times W$ pixels, $\tau$ is a matrix whose rows express each pixel in $\mathcal{E}(T\psi)$ as a linear combination of pixels in $\mathcal{E}(\psi)$, similar to the weight matrix one might obtain from a cross-attention operation.

## 4.1 Regularization Through Isometries

We will find $\tau$ by solving a least-squares problem of the form $\tau = \min_\pi \|\mathcal{E}(T\psi) - \pi \mathcal{E}(\psi)\|$. Unfortunately, as we will show in experiments, a direct solve without additional regularization leads to uninformative maps that capture little information about the actual world-space transformations $T$. To add structure, we might ask that $\tau$ be *orthogonal* with $\tau^\top \tau = I_{|N|}$, generating gradients promoting latent codes having the property $\|\mathcal{E}(\psi)\| \approx \|\mathcal{E}(T\psi)\|$.

However, we can obtain more structure yet. We propose to learn a representation of the latent geometry by jointly regressing a diagonal mass matrix $\mathbb{M}$ and positive semi-definite (**PSD**) operator $\Omega \in \mathbb{R}^{|N| \times |N|}$ such that $\tau$ manifests as an *isometry*. That is, $\tau$ preserves the functional inner product defined by $\mathbb{M}$ – $\langle f, g \rangle_\mathbb{M} = f^\top \mathbb{M} g$ for $f, g \in L^2(N, \mathbb{R}^d)$ – and is $\Omega$-commutative with

$$\tau^\top \mathbb{M} \tau = \mathbb{M} \qquad \text{and} \qquad \tau \Omega = \Omega \tau. \tag{3}$$

Together, the conditions in Equation (3) form a strong regularizer, the effects of which are best seen in the expression of $\tau$ in the eigenspectrum of $\Omega$. As $\Omega$ is a PSD matrix with respect to the inner product defined by $\mathbb{M}$, it can be expressed in terms of its spectral decomposition as

$$\Omega = \Phi \Lambda \Phi^\top \mathbb{M} \qquad \text{with} \qquad \Phi^\top \mathbb{M} \Phi = I_{|N|}, \tag{4}$$

where $\Lambda = \text{diag}(\{\lambda_i\}_{1 \leq i \leq |N|})$ is the diagonal matrix of (non-negative) eigenvalues and $\Phi \in \mathbb{R}^{|N| \times |N|}$ is the matrix whose columns are the $\mathbb{M}$-orthogonal eigenfunctions of $\Omega$. Denoting

$$\tau_\Omega \equiv \Phi^\top \mathbb{M} \tau \Phi \tag{5}$$

as the projection of $\tau$ into the eigenbasis, it can be shown that the conditions in Eq. (3) reduce to $\tau_\Omega$ being orthogonal and $\Lambda$-commutative [27]. This is equivalent to asking that $\tau_\Omega$ be both *sparse* and *condensed* in that it forms an *orthogonal, block-diagonal matrix*, with the size of each block determined by the multiplicity of the eigenvalues in $\Lambda$.

### 4.2 Estimating $\tau$ and End-to-End Optimization

Fig. 2 visualizes Neural Isometries's training loop. First, $\tau$ is estimated between pairs of encoded $T$-related observations such that it approximately satisfies the conditions for an $\Omega$-isometry as in Eq. (3). Second, the weights of the autoencoder, $\mathbb{M}$, and $\Omega$ are jointly updated with respect to a combined loss term, promoting: a) latent equivariance as in Eq. (2), b) the ability of the decoder to reconstruct observations, and c) distinct eigenvalues $\Lambda$ which encourage a diagonal-as-possible $\tau_\Omega$.

**Recovering $\tau$ Between Latents.** Instead of estimating $\tau$ directly, we equivalently estimate $\tau_\Omega$ in the eigenbasis of $\Omega$, motivated by the corresponding simplification of the conditions in Eq. (3). Let

$$\mathcal{E}_\Omega \equiv \Phi^\top \mathbb{M} \circ \mathcal{E} \tag{6}$$

be the map given by encoding followed by projection into the eigenbasis of $\Omega$. Then, given observations $\psi$ and $T\psi$, we define $\tau_\Omega$ to be the solution to the least squares problem

$$\tau_\Omega = \underset{\pi^\top \pi = I, \pi\Lambda = \Lambda\pi}{\text{minimum}} \|\pi \mathcal{E}_\Omega(\psi) - \mathcal{E}_\Omega(T\psi)\|. \tag{7}$$

While Eq. (7) has an exact analytical solution [39], we instead approximate $\tau_\Omega$ with a fuzzy analogue which we find better facilitates backwards gradient flow to the parameters of $\Omega$.

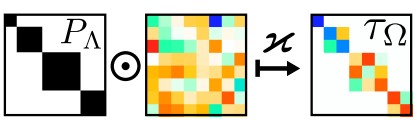

Specifically, letting $\varkappa : \mathbb{R}^{|N| \times |N|} \to \mathrm{O}(|N|)$ denote the Procrustes projection to the nearest orthogonal matrix (*e.g.* through the SVD), we recover $\tau_\Omega$ via the approximation

$$\tau_\Omega \approx \varkappa\left(P_\Lambda \odot \mathcal{E}_\Omega(T\psi)[\mathcal{E}_\Omega(\psi)]^\top\right), \tag{8}$$

where $[P_\Lambda]_{ij} = \exp(-|\lambda_i - \lambda_j|)$ is a smooth multiplicity mask over the eigenvalues $\Lambda$ applied element-wise. See the supplement for details.

To facilitate the recovery of $\tau_\Omega$, we parameterize $\mathbb{M}$ directly by its diagonal elements and $\Omega$ in terms of its spectral decomposition, learning a $\mathbb{M}$-orthogonal matrix of eigenfunctions $\Phi$ and non-negative eigenvalues $\Lambda$. This has the added benefit of enabling a low-rank approximation of $\Omega$ by parameterizing only the first $k$ eigenvalues and eigenfunctions, *i.e.* $\Phi \in \mathbb{R}^{|N| \times k}$ and $\Lambda = \mathrm{diag}(\{\lambda_i\}_{1 \le i \le k})$ with $k \le |N|$, mirroring similar approaches in SoTA FM pipelines [19, 40, 41]. This reduces the complexity of the orthogonal projection in Eq. (8) from $|N| \times |N|$ to $k \times k$.

**Optimization.** During training, the autoencoder is given pairs of $T$-related observations $(\psi, T\psi)$, which are mapped to the latent space and $\tau_\Omega$ is estimated as in Eq. (8) giving $\tau = \Phi \tau_\Omega \Phi^\top \mathbb{M}$. First, an **equivariance loss** is formed between the eigenspace projections of the encodings,

$$\mathcal{L}_E = \|\tau_\Omega \mathcal{E}_\Omega(\psi) - \mathcal{E}_\Omega(T\psi)\|. \tag{9}$$

We note that for full rank $\Omega$, this loss is equivalent to measuring the degree to which the equivariance condition in Eq. (2) holds due to the orthogonality of $\Phi$. Next we compute a **reconstruction loss**,

$$\mathcal{L}_R = \|\mathcal{D}(\tau \mathcal{E}(\psi)) - T\psi\| + \|\mathcal{D}(\tau^{-1} \mathcal{E}(T\psi)) - \psi\|, \tag{10}$$

with $\tau^{-1} = \Phi \tau_\Omega^\top \Phi^\top \mathbb{M}$, forcing the decoder to map the transformed latents to the corresponding $T$-related observations. Last, we formulate a **multiplicity loss** which promotes distinct eigenvalues and ensures $\Omega$ is "interesting" by preventing it from regressing to the identity. We observe that the eigenvalue mask $P_\Lambda$ can be viewed as a graph in which the number of connected components (*i.e.* the number of distinct eigenvalues) is equivalent to the dimension of the nullspace of the graph Laplacian $\Delta_{P_\Lambda}$ formed from the mask [42]. As a measure of the nullspace dimension, we use the norm of the eigenvalues of $\Delta_{P_\Lambda}$, given by $\mathcal{L}_M = \|\Delta_{P_\Lambda}\|$. The NFT [36, 37] takes a similar approach, wherein a diagonalization loss is imposed on estimated transformations themselves, though our experiments show it be far less effective in enforcing structure. The total loss is the sum of aforementioned terms

$$\mathcal{L} = \mathcal{L}_R + \alpha\mathcal{L}_E + \beta\mathcal{L}_M, \tag{11}$$

with $\alpha, \beta \ge 0$ weighting the contributions of the equivariance and multiplicity losses.

In experiments, we also consider a similar triplet regime as proposed in [36, 37], where the autoencoder is given triples of $T$-related observations $(\psi, T\psi, T^2\psi)$ (assuming $T$ is composable) and the estimated map $\tau$ between the encodings of $(\psi, T\psi)$ is used to form equivariance and reconstruction losses between $(T\psi, T^2\psi)$ and vice-versa. This works to prevent $\tau$ from "cheating" by encoding

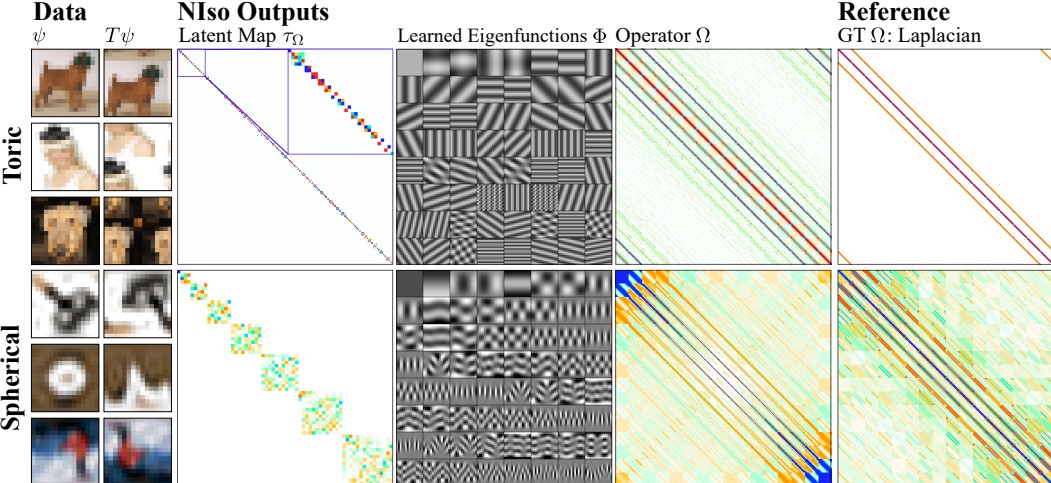

**Data**
$\psi$  $T\psi$  NIso Outputs
Latent Map $\tau_\Omega$   Learned Eigenfunctions $\Phi$ Operator $\Omega$
**Reference**
GT $\Omega$: Laplacian

Toric

Spherical

Figure 3: **Approximating the Laplacian.** Forced to map between shifted images on the torus (first row, left) and rotated images on the sphere (second row, left), NIso regress operators (center right) structurally similar to the toric and spherical Laplacian (right). Maps $\tau_\Omega$ between projected images are strongly diagonal (center left), with individual blocks (inset) preserving the subspaces spanned by eigenfunctions (center, first $64$ shown) sharing nearly the same eigenvalues. These experiments result in the discovery of basis with the similar properties to the the toric and spherical harmonics. In particular, the estimated spherical $\tau_\Omega$ manifest *exactly* the same structure as the ground truth Wigner-D matrices corresponding to the rotation, with square blocks of size $(2\ell + 1) \times (2\ell + 1)$ for the $\ell$-th distinct eigenvalue. Please zoom in to view structural details.

privileged information about the relationship between pairs beyond $T$, a property we show to be critical in enforcing a useful notion of latent equivariance. However, triples of $T$-related observations are rare in practical settings, and we show in experiments that a major benefit of our isometric regularization is that our multiplicity loss (promoting a diagonal-as-possible and thus sparse and condensed $\tau_\Omega$) can serve as an effective substitute for access to triples.

**Spectral Dropout.**  While the composite loss in Equation (11) promotes latent equivariance and distinct eigenvalues, it does not, however, promote a physically meaningful ordering of the eigenvalues – *i.e.* that small eigenvalues correspond to smooth, low-frequency eigenfunctions and larger eigenvalues to eigenfunctions with sharper, high-frequency details. Though such an ordering amounts to a permutation in the spectral dimension and is not a necessary condition for the existence of diagonalized isometric maps, it could be potentially useful in downstream applications where a classical notion of eigenvalues as frequencies is desired.

To this end we propose an *optional* spectral dropout layer applied *after* computing the equivariance loss $\mathcal{L}_E$ and *before* the basis unprojection. The layer is implemented as follows: During training, there is a $50\%$ chance that dropout will be applied to given example in a batch. Then, for each such example, a spectral index $1 < i \le k$ is randomly chosen and all features with index $j \ge i$ are masked out. Intuitively, this forces the decoder to produce the best possible reconstructions with the lowest frequency eigenfunctions, which appear most often and thus must capture large scale features, while reserving higher frequencies for filling in fine details.

### 4.3  A Simple Example: Approximating the Toric and Spherical Laplacians

We demonstrate that NIso are able to learn a compact representation of isometries that reflect the dynamics of transformations in world space. To do so, we perform two experiments in which we consider pairs of observations formed by $16 \times 16$ images from ImageNet [43] viewed functions on toric and spherical grids, and respectively transformed by random circular shifts and SO(3) rotations to form pairs. Thus pairs are related by the isometries of the torus and sphere which commute with the Laplacian on each domain. Taking the encoder and decoder to be the identity map (making the equivariance and reconstruction losses equivalent) we optimize for $\mathbb{M}, \Omega \in \mathbb{R}^{256 \times 256}$ via the pairwise training procedure described in sec. 4.2, including the use of spectral dropout. For the toric experiments, we learn a full-rank parameterization of $\Omega$ with $k = 256$; for the sphere, we learn a

low-rank approximation with $k = 64$. As seen in Fig 3, NIso regresses operators with significant structural similarities to the Laplacian matrices formed by the standard $3 \times 3$ stencil on the torus and the low rank approximation using the first $64$ spherical harmonics. In both cases, NIso recovers an eigenspace that diagonalizes $\tau_\Omega$ between shifted images, with eigenfunctions ordered by their energy. As our approach is data-driven and our estimated maps are only *approximately* isometric, our learned operator and its eigenspectrum do not perfectly correspond to the ground-truth Laplacian. Instead, we are able to characterize similar, non-trivial spatial relationships that are preserved under shifts.

### 4.4 Representation Learning with NIso

Viewed in terms of representation learning, NIso can be seen as a recipe for the self-supervised pre-training of a network backbone $\mathcal{E}$ satisfying the equivariance condition in Eq. (2) such that transformations $T$ in the world space manifest as isometries $\tau$ in the sense of Eq. (3).

**Exploiting Equivariance in Latent Space** As we demonstrate in experiments, a simple off-the-shelf isometry-equivariant head can be appended to the pre-trained backbone and fine-tuned to achieve competitive results in tasks with challenging symmetries. We employ a simple strategy wherein a low rank $k$ approximation of $\Omega$ is learned during the pre-training stage. Thus, the eigenspace projections of the encodings of $T$-related observations are $k \times d$ tensors that are nearly equivalent up to an *orthogonal* transformation $\tau_\Omega$. As such, we pass the projected encodings to a head consisting of an $O(k)$-equivariant vector neuron (**VN**) MLP [38]. We note that for large-scale tasks NIso is potentially well-suited to pair with DiffusionNet [44], which can make use of the learned eigenbasis to perform accelerated operations in the latent space, though we do not consider this regime here.

**Pose Extraction from Latent Isometries.** We propose that the recovered functional maps $\tau$ encode information about world-space transformations $T$. To test this, we consider a simple pose estimation paradigm consisting of a pre-training phase and fine-tuning phase. In the first phase, a NIso autoencoder is trained using $T$-related pairs of observations consisting of adjacent frames from video sequences. Subsequently, the decoder is discarded and the same pairs of observations are considered during fine-tuning. In the second phase, isometries $\tau_\Omega$ are estimated between the eigenspace projections of the encoded observations, vectorized, and passed directly to an MLP which predicts the parameters of the $SE(3)$ transformation corresponding to the relative camera motion in world space between the adjacent frames. In our experiments, the weights of the NIso backbone are frozen during fine-tuning to better evaluate the information about world space transformations encoded during the unsupervised pre-training phase. At evaluation, trajectories are recovered by composing estimated frame-to-frame poses over the length of the sequence.

## 5 Experiments

In this section, we provide empirical evidence through experiments that NIso 1) recovers a general-purpose latent space that can be exploited by isometry-equivariant networks to handle challenging symmetries (5.1, 5.2); and 2) NIso encodes information about transformations in world space through the construction of isometric maps in the latent space from which geometric quantities such as camera poses can be directly regressed (5.3). Here we pre-train NIso without spectral dropout, as the ordering of the learned spectrum is irrelevant in our target applications and we find it slightly decreases accuracy of the prediction head. We provide reproducibility details in the supplement in addition to experiments quantitatively evaluating the degree of learned equivariance. We note that a consistent theme in our experiments are comparisons against the unsupervised variant of the NFT [37] (the semi-supervised variants cannot be applied because the symmetries we consider have no finite-dimensional IRs). Like our approach, the NFT seeks to relate latents via linear transformations though, it differs fundamentally in that maps are guaranteed additional structure only if the world space transformations are a compact group action. While not originally proposed by the authors, we evaluate it in place of our approach in the same self-supervised representation learning regimes discussed in sec. 4.4. Thus the role of these comparisons is to show that our proposed *isometric* regularization better and more consistently provides a tractable and informative latent space.

### 5.1 Homography-Perturbed MNIST

In our first set of experiments, we consider classification on the homNIST dataset [6] consisting of homography-perturbed MNIST digits. Following the procedure outlined in sec. 4.4, the classification

network consists of a pre-trained NIso encoder backbone followed by a VN-MLP. Specifically, pre-training is performed by randomly sampling homographies from the distribution proposed in [6] which are applied to the elements of the standard MNIST training set to create pairs of observations. Here the weights of the encoder backbone are *frozen* and the equivariant head is trained only on the *original* (unperturbed) MNIST training set and evaluated on the *perturbed* test set. Thus the aim of these experiments is to directly quantify the degree to which pre-trained latent space is both equivariant *and* distinguishable.

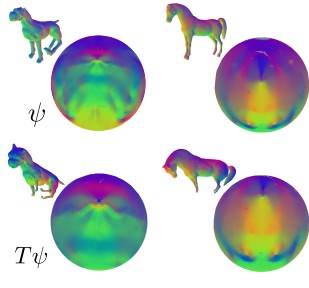

| | Acc. |
|---|---|
| **NIso** | 92.52 ($\pm$ 0.91) |
| w/ triplet | 97.38 ($\pm$ 0.23) |
| w/o $\mathcal{L}_E$ | 77.30 ($\pm$ 2.56) |
| w/o $\mathcal{L}_M$ | 45.27 ($\pm$ 1.20) |
| NFT [37] | 41.93 ($\pm$ 0.84) |
| w/ triplet | 67.15 ($\pm$ 1.10) |
| AE w/ aug. | 80.96 ($\pm$ 1.95) |
| homConv [6] | 95.71 ($\pm$ 0.09) |
| LieDecomp [45] | **98.30 ($\pm$ 0.10)** |

Table 1: **Hom. MNIST.**

With this in mind, we perform three ablations. In the first, we train the NIso autoencoder in a triplet regime (4.2) made possible by the synthetic parameterization of $T$ as homographies. In the second and third, we train the autoencoder in the standard pairwise regime without considering the equivariance loss $\mathcal{L}_E$ and multiplicity loss $\mathcal{L}_M$, respectively. Additionally we compare the efficacy of our approach versus an NFT backbone pre-trained in the same manner. Last, we pre-train and evaluate a baseline backbone which considers only a reconstruction loss without respect to $T$-related pairs, but trains with data-augmentation during the fine-tuning phase by applying randomly sampled homographies.

Results are shown in Tab. 1, averaged over five randomly initialized pre-training and fine-tuning runs with standard errors. Also included are those reported by homConv [6] and LieDecomp [45], top-performing homography-equivariant networks which serve as a handcrafted baseline. While NIso pre-trained with the triplet regime produces results on par with the handcrafted baselines, pre-training with the pairwise regime—which reflects a real-world scenario—achieves a classification accuracy above 90%, significantly better than all but the three aforementioned approaches. Critically, performance drops when the multiplicity loss $\mathcal{L}_M$ is omitted, which corresponds to a regime where $\tau$ must only preserve the inner product and $\Omega$ converges to a multiple of the identity operator. This suggests that sparsifying $\tau_\Omega$ by filtering it through a low-multiplicity eigenspace (*i.e.* enforcing that it is an "interesting" isometry) is fundamental in forcing the network to disentangle the structure of the observed transformations from the content of the observations themselves. In the same vein, the equivariance loss $\mathcal{L}_M$ is also clearly instrumental, as the reconstruction loss alone does not explicitly enforce that $\mathcal{E}(\psi)$ is in fact mapped to $\mathcal{E}(T\psi)$ under the estimated $\tau$. Furthermore, while the authors report that latent maps tend to converge to orthogonal maps for compact group actions in world space [36, 37], both NFT regimes preform poorly, implying that the learned maps do not replicate the properties of finite-dimensional IRs when the group is non-compact.

## 5.2 Conformal Shape Classification

Next, we apply NIso to classify conformally-related 3D shapes from the augmented SHREC '11 dataset [7, 46]. We follow [7] by mapping each mesh to the sphere and subsequently rasterizing to a grid. During pre-training, $T$-related pairs are selected from the sets of conformally-augmented meshes derived from the same base shape in the train split. In the fine-tuning phase, the encoder weights are unfrozen and are jointly optimized with the equivariant head, representing the practical implementation of our proposed approach for equivariant tasks.

| | Acc. |
|---|---|
| **NIso** | **90.26 ($\pm$ 1.27)** |
| NFT [37] | 83.24 ($\pm$ 2.03) |
| AE w/ aug. | 69.36 ($\pm$ 2.81) |
| MC [7] | 86.5 |

Table 2: **Conf. SHREC '11.**

Results are shown in Tab. 2, averaged over five randomly initialized pre-training and fine-tuning runs with standard errors. NIso outperforms the NFT and the autoencoder baseline (with random Möbius transformations applied during the fine-tuning phase) in addition to Möbius Convolutions (**MC**) [7], a SoTA handcrafted spherical network equivariant to Möbius transformations. We consider this dataset to present a significant challenge as shape classes are roughly conformally-related and thus the maps between their spherical parameterizations are only approximated by Möbius transformations.

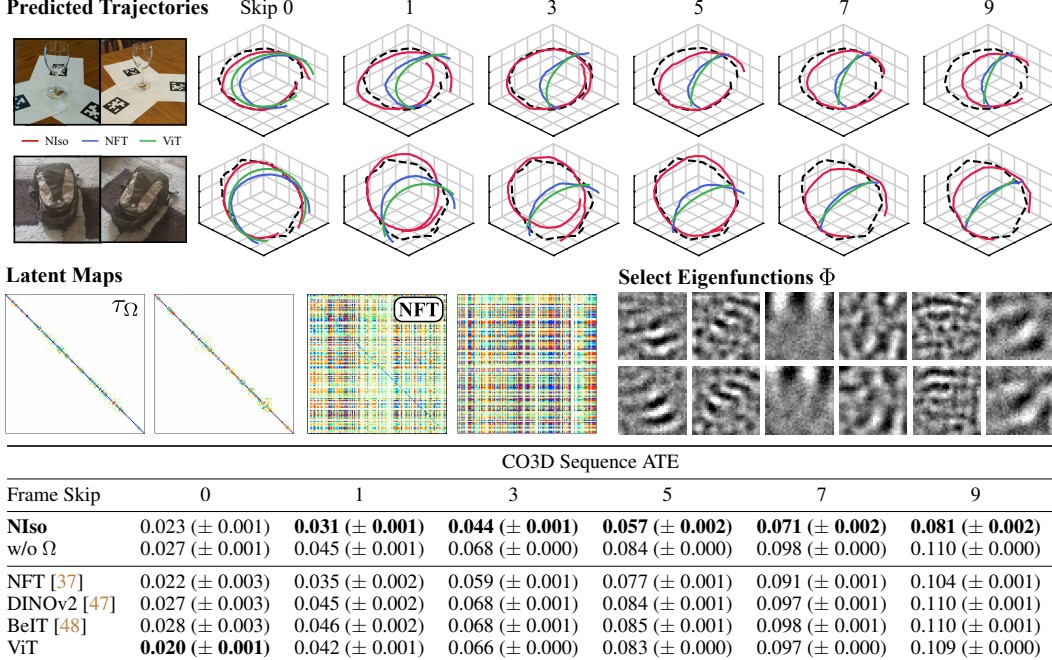

| CO3D Sequence ATE | | | | | | |
|---|---|---|---|---|---|---|
| Frame Skip | 0 | 1 | 3 | 5 | 7 | 9 |
| **NIso** | 0.023 (± 0.001) | **0.031 (± 0.001)** | **0.044 (± 0.001)** | **0.057 (± 0.002)** | **0.071 (± 0.002)** | **0.081 (± 0.002)** |
| w/o $\Omega$ | 0.027 (± 0.001) | 0.045 (± 0.001) | 0.068 (± 0.000) | 0.084 (± 0.000) | 0.098 (± 0.000) | 0.110 (± 0.000) |
| NFT [37] | 0.022 (± 0.003) | 0.035 (± 0.002) | 0.059 (± 0.001) | 0.077 (± 0.001) | 0.091 (± 0.001) | 0.104 (± 0.001) |
| DINOv2 [47] | 0.027 (± 0.003) | 0.045 (± 0.002) | 0.068 (± 0.001) | 0.084 (± 0.001) | 0.097 (± 0.001) | 0.110 (± 0.001) |
| BeIT [48] | 0.028 (± 0.003) | 0.046 (± 0.002) | 0.068 (± 0.001) | 0.085 (± 0.001) | 0.098 (± 0.001) | 0.110 (± 0.001) |
| ViT | **0.020 (± 0.001)** | 0.042 (± 0.001) | 0.066 (± 0.000) | 0.083 (± 0.000) | 0.097 (± 0.000) | 0.109 (± 0.000) |

Table 3: **Pose Estimation Comparison.** Mean ATE for each method across all frame skips on CO3Dv2 evaluation sequences. *Top:* Examples of predicted trajectories, with ground truth in black, NIso in red, the NFT in blue, and the transformer baseline in green. Additional examples are shown in the supplement. *Center Left:* Representative examples of latent maps formed by NIso and the NFT, with the latter shown after applying the authors' proposed diagonalization procedure. NIso is able to form highly sparse, block diagonal maps. In contrast, the NFT struggles to form similarly condensed maps. *Center Right:* Selected learned eigenfunctions. Each column forms a subspace, reflecting fundamental modes of symmetry identified by NIso.

This makes the performance of NIso particularly notable as it suggests that our framework has the potential to offer a more flexible and effective alternative than specialized, handcrafted equivariant networks whose underlying group-based architectures, though elegant, can only approximate inexact symmetries. Furthermore, while the NFT is relatively more competitive with the encoder backbone unfrozen, its performance and that of the baseline indicate that a lack of existing equivariant structure cannot easily be overcome in the fine-tuning phase.

## 5.3 Camera Pose Estimation from Real-World Video

Last, we apply NIso to the task of camera pose estimation from real-world video on the CO3Dv2 dataset [49] following the procedure described in sec. 4.4. Due to the varying quality of the ground-truth trajectories in the dataset, we create train and test sets from the top 25% of sequences in the dataset as ranked by the provided pose quality scores. In particular, we are interested in the general ability of our method to encode information about both small and large scale world space transformations. Thus we train and evaluate in a variable baseline regime, randomly skipping between 0 and 9 frames between pairs during the pre-training and fine-tuning phases. Evaluation is performed by computing the mean absolute trajectory error (**ATE**) between the ground truth trajectories and those recovered by our method over the sequences in the evaluation set. To understand the efficacy of our approach at different scales, we report the mean ATE over six splits consisting of the same sequences in the evaluation set with frame skips of 0, 1, 3, 5, 7, and 9.

We compare against the NFT using the same pre-training and fine-tuning procedure along with a transformer baseline. The latter is inspired by the recent success of such models in 3D vision tasks [50, 51], and uses the same ViT-based architecture proposed in DUSt3R [51], with the decoder modified to directly predict the parameters of the relative camera pose. As the weights of the NIso and NFT backbones are frozen during the fine-tuning phase, the transformer baseline is trained from scratch in the fine-turning phase to more directly compare the descriptiveness of the latent representations relative to the information contained in the raw observations. We also compare

against two strong representation learning baselines. We extract features from both images using two pre-trained state-of-the-art vision foundation models — DINOv2[47] and BeIT[48] — and pass the tokens into the modified DUSt3R-style decoder which is trained to predict the pose.

We additionally train and evaluate an ablative version of NIso which does not learn an operator. Here $\tau$ is predicted directly between encodings and is required only to be orthogonal with the equivariance and reconstruction losses alone enforced during training. We note that computing a block-diagonalization loss on $\tau$ directly lacks justification as it would promote maps that preserve contiguous chunks of spatial indices in the latent tensors which are ordered arbitrarily. Thus, this regime serves to evaluate the degree to which the regularization through learned $\Omega$-commutativity forces the latent transformations to reflect geometric relationships in world space.

Results are shown in Tab. 3, averaged over five randomly initialized pre-training and fine-tuning runs with standard errors. All methods perform similarly with 0 frame skip but diverge afterwards, with NIso achieving significantly lower ATE values as the skip length increases. Notably, the ablative version of our method is consistently among the worst performers, suggesting that isometric regularization is also critical to encode information about world space transformations. Overall the NFT achieves the second best performance, slightly outperforming NIso at 0 frame skip but diverging thereafter. However, the latent maps it recovers are neither sparse nor exhibit condensed structure (Tab. 3, center left), and we hypothesize that its inability to effectively regularize maps beyond linearity makes it difficult for the network to discover a consistent transformation model that applies across scales. This could cause the network to focus on a specific regime at the expense of others, which may explain its relatively strong performance at 0 frame skip. The transformer baseline and representation learners are also highly flexible, and an analogous line of reasoning could explain their similar error profile.

## 6    Discussion

**Limitations.**    A key factor limiting the broader applicability of our approach to geometry processing and graph-based tasks is an inability to learn and transfer an operator between domains with varying connectivity. In addition, NIso does not explicitly handle partiality or occlusion between input pairs which are ubiquitous in real-world data and likely degrade its performance in the pose estimation task. We seek to address these limitations in future work.

**Conclusion.**    In this paper we introduce Neural Isometries, a method which converts challenging observed symmetries into isometries in the latent space. Our approach forms an effective backbone for self-supervised representation learning, enabling simple off-the-shelf equivariant networks to achieve strong results in tasks with complex, non-linear symmetries. Furthermore, isometric regularization produces latent representations that are geometrically informative by encoding information about transformations in world space, and we demonstrate that robust camera poses can be extracted from the isometric maps between latents in a general baseline setting.

**Acknowledgements.**    Use of the CO3Dv2 dataset is solely for benchmarking purposes and limited exclusively to the experiments described in sec. 5.3. We thank Ishaan Chandratreya and Hyunwoo Ryu for helpful discussions and David Charatan for his aesthetic oversight and for granting us permission to use his images captured for FlowMap [52] in Fig. 1-2.

Vincent Sitzmann was supported by the National Science Foundation under Grant No. 2211259, by the Singapore DSTA under DST00OECI20300823 (New Representations for Vision and 3D Self-Supervised Learning for Label-Efficient Vision), by the Intelligence Advanced Research Projects Activity (IARPA) via Department of Interior/ Interior Business Center (DOI/IBC) under 140D0423C0075, by the Amazon Science Hub, and by IBM.

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

# A Derivations and Implementation Details

## A.1 Isometries in the Eigenbasis

Here we show that isometries manifest as orthogonal matrices that commute with the diagonal matrix of eigenvalues in the operator eigenbasis. Suppose we are given a PSD operator $\Omega$ and diagonal mass matrix $\mathbb{M}$, with the former expressed in terms of its eigendecomposition as in Equation (4) for eigenfunctions $\Phi$ and eigenvalues $\Lambda$.

Let $\tau$ be an isometric functional map satisfying the conditions in Equation (3) with $\tau_\Omega$ its projection into the eigenbasis as in Equation (5). Noting that the condition $\Phi^\top \mathbb{M} \Phi = I_{|N|}$ implies that $\Phi^{-1} = \Phi^\top \mathbb{M}$ and $\Phi^{-\top} = \mathbb{M}\Phi$, it follows that

$$
\begin{aligned}
\tau_\Omega{}^\top \tau_\Omega &= [\Phi^\top \tau^\top \, \mathbb{M} \, \Phi] \Phi^\top \mathbb{M} \, \tau \, \Phi \\
&= \Phi^\top \tau^\top \mathbb{M} \, \tau \, \Phi \\
&= \Phi^\top \mathbb{M} \Phi = I_{|N|},
\end{aligned}
$$

and thus $\tau_\Omega$ is orthogonal. Furthermore, it follows that

$$
\begin{aligned}
\tau_\Omega \Lambda &= \Phi^\top \mathbb{M} \, \tau \, \Phi \Lambda \\
&= \Phi^\top \mathbb{M} \, \tau \, \Phi \Lambda \Phi^\top \mathbb{M} \Phi \\
&= \Phi^\top \mathbb{M} \, \tau \, \Omega \Phi \\
&= \Phi^\top \mathbb{M} \Omega \, \tau \, \Phi \\
&= \Lambda \Phi^\top \mathbb{M} \, \tau \, \Phi = \Lambda \tau_\Omega,
\end{aligned}
$$

so $\tau_\Omega \Lambda = \Lambda \tau_\Omega$.

## A.2 Recovering $\tau_\Omega$

Here we derive the approximate solution to the least squares problem in Equations (7 – 8). Specifically, let $\Lambda = \operatorname{diag}(\{\lambda_i\}_{1 \le i \le k}) \in \mathbb{R}^{k \times k}$ be a diagonal matrix of non-negative eigenvalues with $\Lambda_{ii} \le \Lambda_{i+1,i+1}$. Furthermore, suppose that there are $q$ distinct eigenvalues $\{\widetilde{\lambda}_i\}_{1 \le i \le q}$, $\widetilde{\lambda}_i \le \widetilde{\lambda}_{i+1}$ each with multiplicity $m_i$. Thus, $\Lambda$ can be expressed as the direct sum

$$
\Lambda = \bigoplus_{i=1}^{q} \widetilde{\lambda}_i I_{m_i}, \tag{12}
$$

with $I_{m_i}$ denoting the $m_i \times m_i$ identity matrix. It follows that any $k \times k$ matrix $\pi$ satisfying $\pi\Lambda = \Lambda\pi$ must be of the form

$$
\pi = \bigoplus_{i=1}^{q} \pi_i, \qquad \pi_i \in \mathbb{R}^{m_i \times m_i}, \ 1 \le i \le q. \tag{13}
$$

That is, $\pi$ must be a block diagonal matrix with the size of each block determined by the multiplicity of the eigenvalues of $\Lambda$.

Now, given any $A, B \in \mathbb{R}^{k \times d}$ consider the minimization problem

$$
\pi^* = \underset{\pi^\top \pi = I, \pi\Lambda = \Lambda\pi}{\operatorname{minimum}} \|\pi A - B\|, \tag{14}
$$

consisting of finding an orthogonal, $\Lambda$-commuting map minimizing the distance between $A$ and $B$ under the Frobenius norm. Writing

$$
A = \begin{bmatrix} A_1 \\ \vdots \\ A_q \end{bmatrix} \quad \text{and} \quad B = \begin{bmatrix} B_1 \\ \vdots \\ B_q \end{bmatrix}, \tag{15}
$$

the solution to the minimization problem in Equation (14) is equivalently expressed as the direct sum of the solutions to the orthogonal sub-problems corresponding to each block [39] with

$$
\pi^* = \bigoplus_{i=1}^{q} \underset{\pi_i^\top \pi_i = I}{\operatorname{minimum}} \|\pi_i A_i - B_i\|. \tag{16}
$$

Now, for arbitrary $n$ let $\varkappa : \mathbb{R}^{n \times n} \to \mathrm{O}(n)$ denote the Procrustes projection to the nearest orthogonal matrix such that for any $M \in \mathbb{R}^{n \times n}$ with SVD $M = U\Sigma V^\top$, $\varkappa(M) \equiv UV^\top$. Thus, the solution to the minimization problem is given by

$$\pi^* = \bigoplus_{i=1}^{q} \varkappa(B_i A_i^\top), \tag{17}$$

which, following from the properties of the SVD, can be expressed equivalently via a single $k \times k$ Procrustes projection such that

$$\pi^* = \varkappa\left( \bigoplus_{i=1}^{q} B_i A_i^\top \right). \tag{18}$$

Here we observe that by defining $P_\Lambda \in \mathbb{R}^{k \times k}$ to be the eigenvalue mask given by

$$[P_\Lambda]_{ij} = \begin{cases} 1, & \lambda_i = \lambda_j \\ 0, & \text{otherwise} \end{cases} \tag{19}$$

we have

$$P_\Lambda \odot BA^\top = \bigoplus_{i=1}^{q} B_i A_i^\top, \tag{20}$$

and thus

$$\pi^* = \varkappa(P_\Lambda \odot BA^\top). \tag{21}$$

In practice we substitute $P_\Lambda$ in Equation (19) with the fuzzy analogue $[P_\Lambda]_{ij} = \exp(-|\lambda_i - \lambda_j|)$, which we find better facilitates the flow of backwards gradients to the parameters of $\Omega$. Replacing $A$ and $B$ with $\mathcal{E}_\Omega(\psi)$ and $\mathcal{E}_\Omega(T\psi)$ we arrive at the approximate solution for $\tau_\Omega$ in Equation (8).

### A.3 Graph Laplacian of $P_\Lambda$

Given the $k \times k$ eigenvalue mask $P_\Lambda$ with $[P_\Lambda]_{ij} = \exp(-|\lambda_i - \lambda_j|)$, we form its graph Laplacian as

$$\Delta_{P_\Lambda} \equiv \mathrm{diag}(P_\Lambda \mathbf{1}) - P_\Lambda, \tag{22}$$

with $\mathbf{1} \in \mathbb{R}^k$ the vector of ones.

## B  Reproducibility

Code and experiments are available at https://github.com/vsitzmann/neural-isometries.

### B.1  Hardware

All experiments were performed on a single NVIDIA A6000 GPU with 48 GB of memory.

### B.2  Approximating the Laplacian

In the toric/spherical experiments, the model consists of a single NIso layer. which learns a full-rank/low-rank operator and mass matrix, with the former parameterized by an orthogonal matrix of eigenvectors $\Phi \in \mathbb{R}^{256 \times 256}/\Phi \in \mathbb{R}^{256 \times 64}$ and diagonal matrix of non-negative eigenvalues $\Lambda$. During training, input features are created by stacking 86 $16 \times 16 \times 3$ examples from ImageNet into a single $16 \times 16 \times 258$ tensor representing an observation $\psi$. In the toric experiments, random circular shifts about both spatial axes are applied to form $T\psi$. In the spherical experiments, the images are assumed to lie on a $16 \times 16$ Driscoll-Healy spherical grid and random rotations in $\mathrm{SO}(3)$ are sampled and applied to form $T\psi$. As the encoder and decoders are the identity maps, the equivariance and reconstruction losses are equivalent, and the objective in Equation (11) is minimized with $\alpha = 0$ and $\beta = 0.1$. The models is trained with spectral dropout for 100,000 iterations with a batch size of 1 using the AdamW optimizer [53] with a weight decay of $10^{-4}$. The learning rate follows a schedule consisting of a 2,000 step warm up from 0.0 to $5 \times 10^{-4}$ and afterwards decays to $5 \times 10^{-5}$ via cosine annealing. Training takes approximately 6 hours.

## B.3 Homography-Perturbed MNIST

**Architecture.** The encoder and decoder are mirrored 2-level ConvNets. Specifically, each level consists of three convolutional ResNet blocks with 128 and 256 channels at the finest and coarsest resolutions respectively. A mean pool (nearest neighbor unpool) layer halving (doubling) the spatial resolution bridges the two levels.

The latent space has 32 dimensions. A rank $k = 32$ operator $\Omega$ and mass matrix $\mathbb{M}$ are learned, with the former parameterized by 32 eigenfunctions and non-negative eigenvalues.

The equivariant classification head consists of a 2-layer VN-MLP [38] with 128 channels, followed by an output layer which computes invariant features from the inner products between vectors which are passed to a standard linear layer to output class predictions.

**Pre-Training.** Following [6], MNIST digits are padded to $40 \times 40$, and pairs are produced by warping digits with respect to homographies $T$ sampled from the proposed distribution. In the triplet regimes, tuples are created by applying $T^2$ in addition to $T$. Both the pairwise and triplet regimes are trained with respect to the composite loss with $\alpha = 0.5$ and $\beta = 0.1$. The two ablative regimes are trained with $\alpha = 0$ and $\beta = 0$, respectively. In the triplet regime, denoting $\tau$ and $\sigma$ to be the maps estimated between $(\mathcal{E}(\psi), \mathcal{E}(T\psi))$ and $(\mathcal{E}(T\psi), \mathcal{E}(T^2\psi))$ respectively, the equivariance and reconstruction losses are formulated as

$$\mathcal{L}_E = \|\sigma_\Omega \, \mathcal{E}_\Omega(\psi) - \mathcal{E}_\Omega(T\psi)\| + \|\tau_\Omega \, \mathcal{E}_\Omega(T\psi) - \mathcal{E}_\Omega(T^2\psi)\| \tag{23}$$

and

$$\mathcal{L}_R = \|\mathcal{D}(\sigma \, \mathcal{E}(\psi)) - T\psi\| + \|\mathcal{D}(\tau \, \mathcal{E}(T\psi)) - T^2\psi\|. \tag{24}$$

The autoencoders are trained without spectral dropout for $50,000$ steps with a batch size of 16 using the AdamW optimizer with a weight decay of $10^{-4}$. The learning rate follows a schedule consisting of a 2,000 step warm up from $0.0$ to $5 \times 10^{-4}$ and afterwards decays to $5 \times 10^{-5}$ via cosine annealing. Training takes approximately 30 minutes.

**Fine-Tuning.** During the fine tuning phase, the decoder is discarded, and the weights of the encoder, operator $\Omega$, and mass $\mathbb{M}$ are frozen. Examples $\psi$ are encoded, and their eigenspace projections $\mathcal{E}_\Omega$ are passed to the classification head to form class predictions under a standard softmax cross-entropy loss. Training is performed for 10,000 iterations with a batch size of 16 using the AdamW optimizer the same weight decay and learning rate schedule as used previously. Training takes approximately one minute. Evaluation is performed on the fixed test set proposed in [6].

**Baselines.** We implement the NFT following the procedure described in the paper [37]. While a ViT-based encoder is proposed in the original paper, we find that it produces worse evaluation results than the 2-layer ConvNet encoder described above and thus use the latter in our evaluations to provide a fair comparison. We follow the authors' proposed implementation and consider only a reconstruction loss and learn a diagonalization transform in a post-processing step. Otherwise, the NFT is pre-trained, fine-tuned, and evaluated identically to our own approach, and uses the same equivariant classification head. We note that the dimension of the latent map constructed by the NFT is the same as $\tau_\Omega$ ($32 \times 32$) and that the tensor passed to the classification head is the same size for both our method and the NFT.

The autoencoder baseline consists of the same encoder and decoder, pre-trained identically but with respect to the standard reconstruction loss $\|\mathcal{D}(\mathcal{E}(\psi)) - \psi\|$ without considering $T$-related pairs. During the fine-tuning phase, we apply data augmentation to the input images by transforming them by homographies sampled from the distribution proposed in [6] before encoding. Latents are passed directly to a standard non-equivariant 2-layer MLP with 128 channels to form class predictions. Otherwise, fine-tuning and evaluation are performed in the same manner.

## B.4 Conformal Shape Classification

**Dataset.** Experiments are performed using the conformally extended SHREC '11 shape classification dataset proposed in [7], with 30 random conformal transformations are applied to each shape to extend the original dataset [46]. The augmented dataset contains 25 distinct shape classes. Each shape is mapped to the sphere using the method of [54]. Input features are taken to be the values of the heat kernel signature [55] computed on the original mesh at 16 timescales logarithmically distributed in the

range $[-2, 0]$ and rasterized to $96 \times 192$ spherical grids. Pairs of $T$-related observations are formed by randomly selecting two conformal augmentations of the same shape. For each set of training and fine-tuning runs, train and evaluation splits are randomly generated by selecting respectively 10 and 4 of the 20 sets of conformally augmented shapes per class.

**Architecture.** The encoder and decoder are mirrored 4-level ConvNets, with two ResNet blocks per level and 32, 64, 128, 256 channels per layer from finest to coarsest resolution. Mean pooling and nearest neighbor upsampling are used to halve and double the resolution between layers.

The latent space has 64 dimensions. A rank $k = 64$ operator $\Omega$ and mass matrix $\mathbb{M}$ are learned, with the former parameterized by 64 eigenfunctions and non-negative eigenvalues.

The equivariant classification head consists of a 4-layer VN-MLP with 128 channels, using the same output layer as described in sec. (B.3) to produce class predictions.

**Pre-Training.** Training is performed with respect to the composite loss with $\alpha = 0.5$ and $\beta = 0.01$. The model is trained without spectral dropout for 50,000 iterations with a batch size of 16 using the AdamW optimizer with a weight decay of $10^{-4}$. The learning rate follows a schedule consisting of a 2,000 step warm up from $0.0$ to $5 \times 10^{-4}$ and afterwards decays to $5 \times 10^{-5}$ via cosine annealing. Training takes approximately four hours.

**Fine-Tuning.** During the fine tuning phase, the decoder is discarded and the encoder, operator $\Omega$, and mass matrix $\mathbb{M}$ are jointly optimized with the equivariant classification head using the standard softmax cross-entropy loss. Training is performed for 10,000 iterations with a batch size of 16 using the AdamW optimizer, with the same weight decay and learning rate schedule as used previously. Training takes approximately one hour.

**Baselines.** We pre-train, fine-tune, and evaluate the NFT and autoencoder baseline using the same autoencoder architecture, classification head, and training regimes in the same manner as sec. (B.3). Here data augmentation is applied during the fine-tuning phase by randomly sampling and applying Möbius transformations with scale factors proportional to those estimated between the spherical parameterizations of the shapes. Again, the both the size of the latent map and size of the tensor passed to the classification are the same size for both our method and the NFT. Here, the weights of the encoder are left unfrozen, and are jointly optimized with the classification head during the fine-tuning phase.

### B.5 Camera Pose Estimation from Real-World Video

**Dataset.** Experiments are performed with a subsection of the CO3Dv2 dataset [49]. Given the high-degree of variability in the quality of the ground truth trajectories, we consider only the top 25% of sequences in the dataset as ranked by the provided pose quality score, slightly under 9,000 trajectories in total. An evaluation set is formed by withholding 10% of said trajectories, with the rest used for training. The list sequences in the train and evaluation sets will be made available along with the code release. As the videos in the dataset are of different resolutions, we center crop each frame to the minimum of its height and width dimensions and resize to $144 \times 144$.

**Architecture.** The encoder and decoder are mirrored 3-level ConvNets, with four ResNet blocks per level and 64, 128, 256 channels per layer from finest to coarsest resolution. Mean pooling and nearest neighbor resolution are used to halve and double the resolution between layers.

The latent space has 128 dimensions. A rank $k = 128$ operator $\Omega$ parameterized by and mass matrix $\mathbb{M}$ are learned, with the former parameterized by 128 eigenvectors and eigenvalues.

The pose extraction network consists of a standard 5-layer MLP with 512 channels, which outputs 9-dimensional tensor representing the parameters of an SE(3) transformation (translation + two vectors used to form the rotation matrix through cross products).

**Pre-Training.** During pre-training, $T$-related pairs of observations are formed by randomly selecting adjacent frames from sequences with a frame skip between 0 and 10. Training is performed with respect to the composite loss with $\alpha = 0.5$ and $\beta = 0.025$. The model is trained for 200,000 iterations with a batch size of 8 using the AdamW optmizer with a weight decay of $10^{-4}$. The learning rate follows a schedule consisting of a 2,000 step warm up from $0.0$ to $5 \times 10^{-4}$ and afterwards decays to $5 \times 10^{-5}$ via cosine annealing. Training takes approximately 12 hours.

**Fine-Tuning.** During the fine-tuning phase, the weights of the encoder, operator $\Omega$, and mass matrix $\mathbb{M}$ are frozen. As in the pre-training phase, pairs of observations are passed to the decoder and $\tau_\Omega \in \mathbb{R}^{128 \times 128}$ is estimated between the eigenspace projection of the encodings. Then, $\tau_\Omega$ is vectorized and passed to the MLP head to predict the parameters of the camera pose.

Training is performed with respect to a two term loss measuring the degree to which the predicted translation and rotation deviate from those of the ground truth pose. Denoting $R_\Omega$ and $R$ to be the predicted and ground truth poses respectively, the rotational component of the loss is given by

$$\mathcal{L}_R = \|R - R_\Omega\|. \tag{25}$$

The transitional component of the loss must be scale invariant as the depth scale factor is unknown between sequences. To handle this, we consider sub-batches formed from pairs of frames *all* belonging to the same sequence. Denoting $t_\Omega$ and $t$ to be the matrices consisting of the sub-batched predicted and ground truth translations stacked column-wise, we form the translational component of the loss via

$$\mathcal{L}_T = \|t - s \cdot t_\Omega\|, \tag{26}$$

where

$$s = \frac{\text{tr}(t^\top t_\Omega)}{\|t_\Omega\|^2} \tag{27}$$

is the scale factor that minimizes $\mathcal{L}_T$ over the pairs of frames in the sub-batch. The total loss is given by $\mathcal{L} = \mathcal{L}_R + \mathcal{L}_T$

The prediction head is trained without spectral dropout for 20,000 iterations with a batch size of 16 (consisting of 4 sub-batches of pairs of frames from 4 sequences) using the AdamW optimizer with the same weight decay and learning rate schedule as used previously. Training takes approximately one 80 minutes.

**Baselines.** The NFT is pre-trained, fine-tuned, and evaluated using the same autoencoder architecture, prediction head, and training regime as our own method in the same manner as is done in the preceding experiments. We note that both our method and the NFT construct latent maps of the same dimension, which are passed to identical prediction heads.

The transformer baseline is based on the CroCo/DUSt3R[50, 51] architectures. As with our approach, input frames are passed to the same decoder. Here the encoder consists of 12-layer VITs with a patch size of 16, with 16 heads and 512 channels per layer. Adjacent frames are independently encoded, and then concatenated along the spatial dimension in addition to a single learned token. The resulting tensor is passed to a 12-layer transformer decoder which mirrors the encoder with 16 heads and 512 channels per layer. Subsequently, the learned token is extracted from the output and passed through a linear layer to recover the predicted pose parameters.

Additionally, we also evaluate the efficacy of DINOv2[47] and BeIT[48] as representation learners for the pose estimation task. Pairs of adjacent frames are passed through the pre-trained models and the feature tokens for each image from the final layers are extracted. These are used as input to the same DUSt3R[47]-style decoder described above which is trained to predict the parameters of the relative camera pose.

The VIT model and DINOv2/BeIT prediction heads are trained from scratch in the fine-tuning phase, using the same loss and optimization procedure as described above.

## C Additional Results

Below we provide additional results including a quantiative evaluation of learned equivariance, visualizations of the eigenfunctions $\Phi$ and mass matrices $\mathbb{M}$ learned in each experiment, and comparisons of estimated camera trajectories recovered by each method in the pose prediction experiments. For the latter, examples of failure cases are also included.

|  | Equivariance Error | |
|  | Hom. MNIST | Conf. SHREC '11 |
| --- | --- | --- |
| **NIso** | 4.06% (± 3.79) | 5.91% (± 3.49) |
| homConv [6] | 6.86% | — |
| MobiusConv [7] | — | 2.64% |

| CO3D Equivariance Error | | | | | | |
| --- | --- | --- | --- | --- | --- | --- |
| Frame Skip | 0 | 1 | 3 | 5 | 7 | 9 |
| NIso | 5.25% (± 1.61) | 5.84% (± 1.41) | 7.44% (± 1.43) | 8.23% (± 1.48) | 8.79 % (± 1.56) | 9.24% (± 1.67) |

Table 4: **Quantitative Evaluation of Learned Equivariance.** Mean NIso equivariance error in the latent space across all experiments. Following [6, 7, 45], we measure the learned equivariance of the NIso latent space in the standard way via the average of $\|\tau_\Omega \, \mathcal{E}_\Omega(\psi) - \mathcal{E}_\Omega(T\psi)\|^2 / \|\mathcal{E}_\Omega(T\psi)\|^2$ over all pairs in the test set, for five randomly initialized pre-training runs. For Hom. MNIST, we randomly sample homographies from the distribution proposed in [6] which are applied to the elements of the standard MNIST test set; for Conf. SHREC '11, pairs are formed from the sets of conformally-augmented meshes derived from the same base shape in the test split. With CO3D we measure the error between encodings of adjacent frames in test set, with increasing frame skip. We also list the errors reported by competing hand-crafted methods.

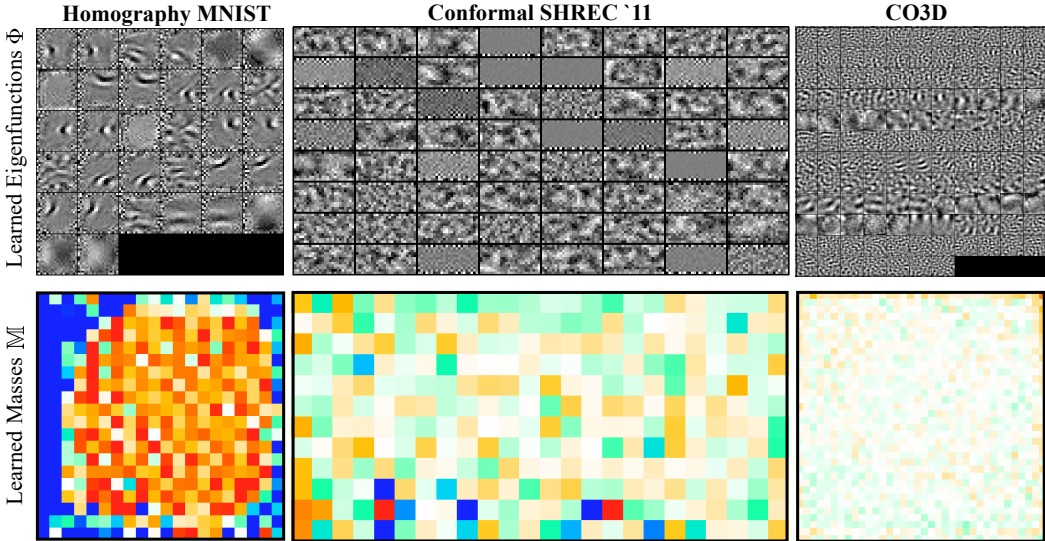

Figure 4: **Visualizing the Learned Eigenfunctions $\Phi$ and Mass Matrices $\mathbb{M}$.** Visualizations of the eigenfunctions $\Phi$ learned in each experiment are shown on the top row. Eigenfunctions are sorted by eigenvalue in ascending order along rows in C-style indexing. Here, experiments were performed without spectral dropout so the ordering is random. The elements of the learned diagonal mass matrices $\mathbb{M}$ are shown on the bottom row, in terms of the magnitude of the deviation from the mean value at each grid index in the latent space. White indicates little deviation from the mean, with green-blue indicating mass values above the mean and orange-red indicating mass values below. In the MNIST experiments (sec. 5.1), the distribution of mass appears to segment null space from the central region most often occupied by the digits. In the conformal shape classification experiments (sec. 5.2), the larger deviations from the mean values appear closer to the poles (the top-most and bottom-most rows of the spherical grid). For the pose estimation experiments (sec. 5.3), larger deviations appear at the boundaries, with the lower half of the grid having slightly higher values.

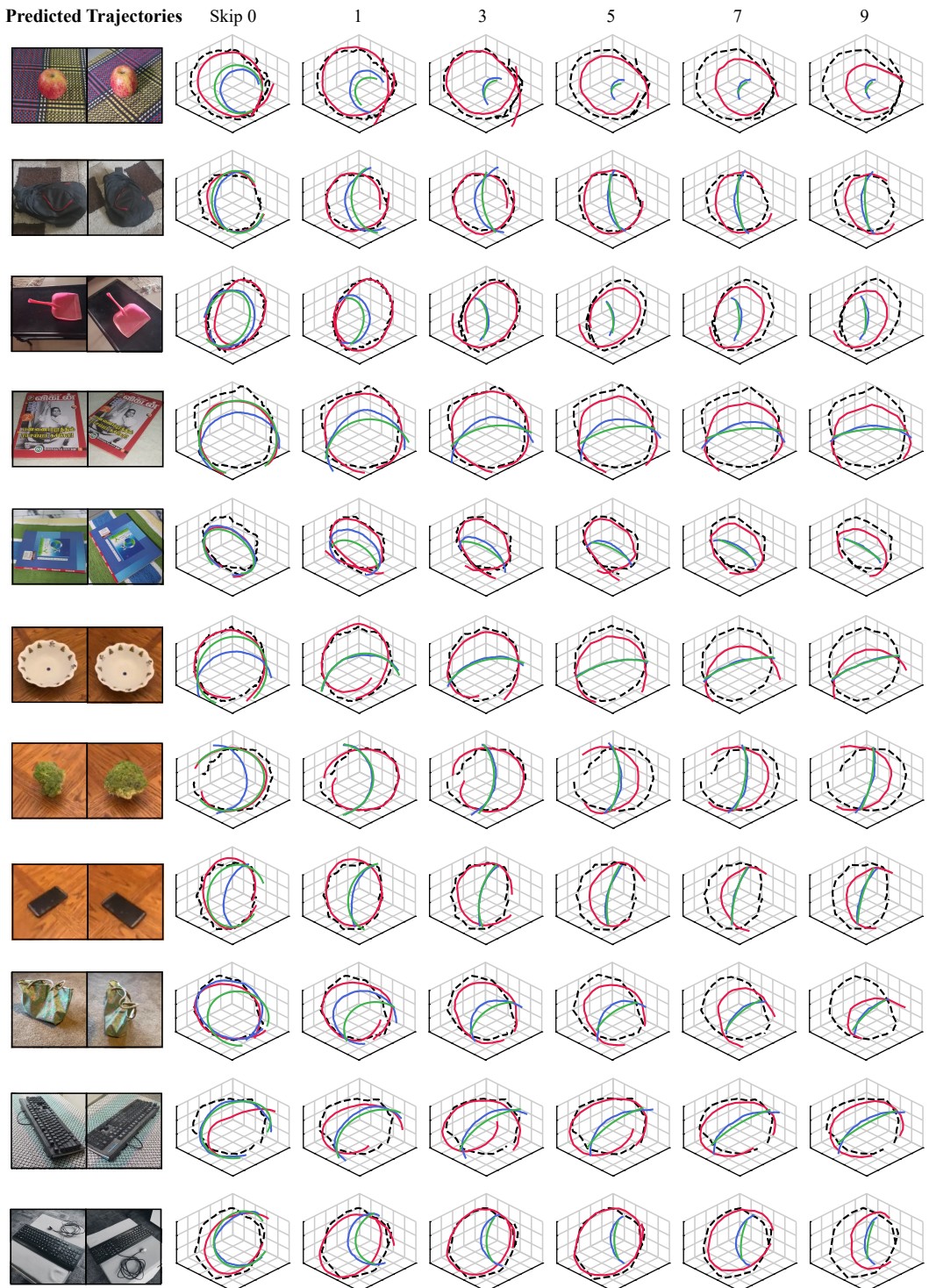

Figure 5: **Qualitative Pose Estimation Comparisons.** Example predicted trajectories for each method on select CO3Dv2 evaluation sequences. Ground truth is shown in black, NIso in red, the NFT in blue, and the transformer baseline in green. NIso appears to consistently better capture rotational (curvature) information about the world space transformations, helping it to better track the the camera motion across scales.

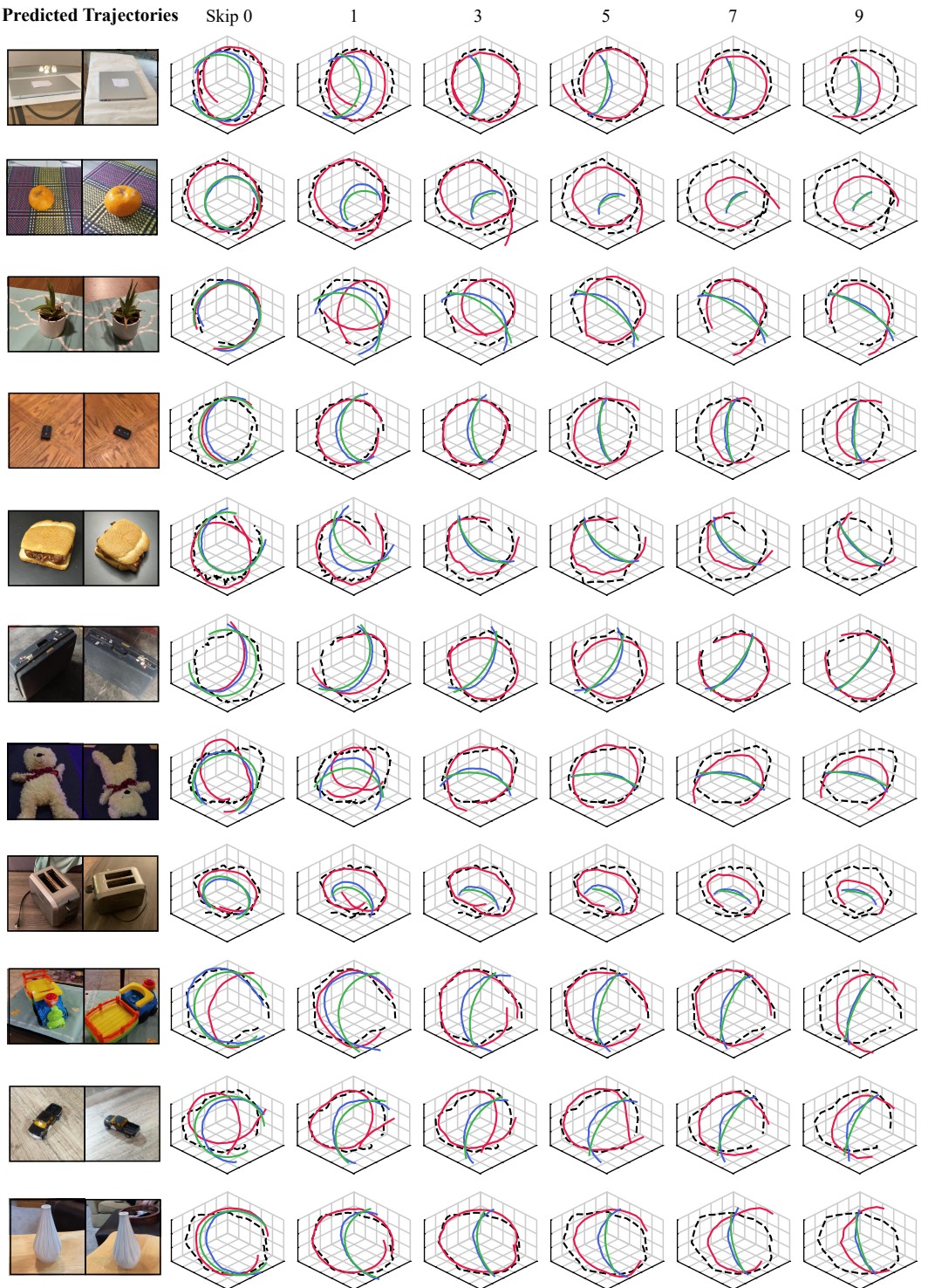

Figure 6: **Qualitative Pose Estimation Comparisons Cont.**

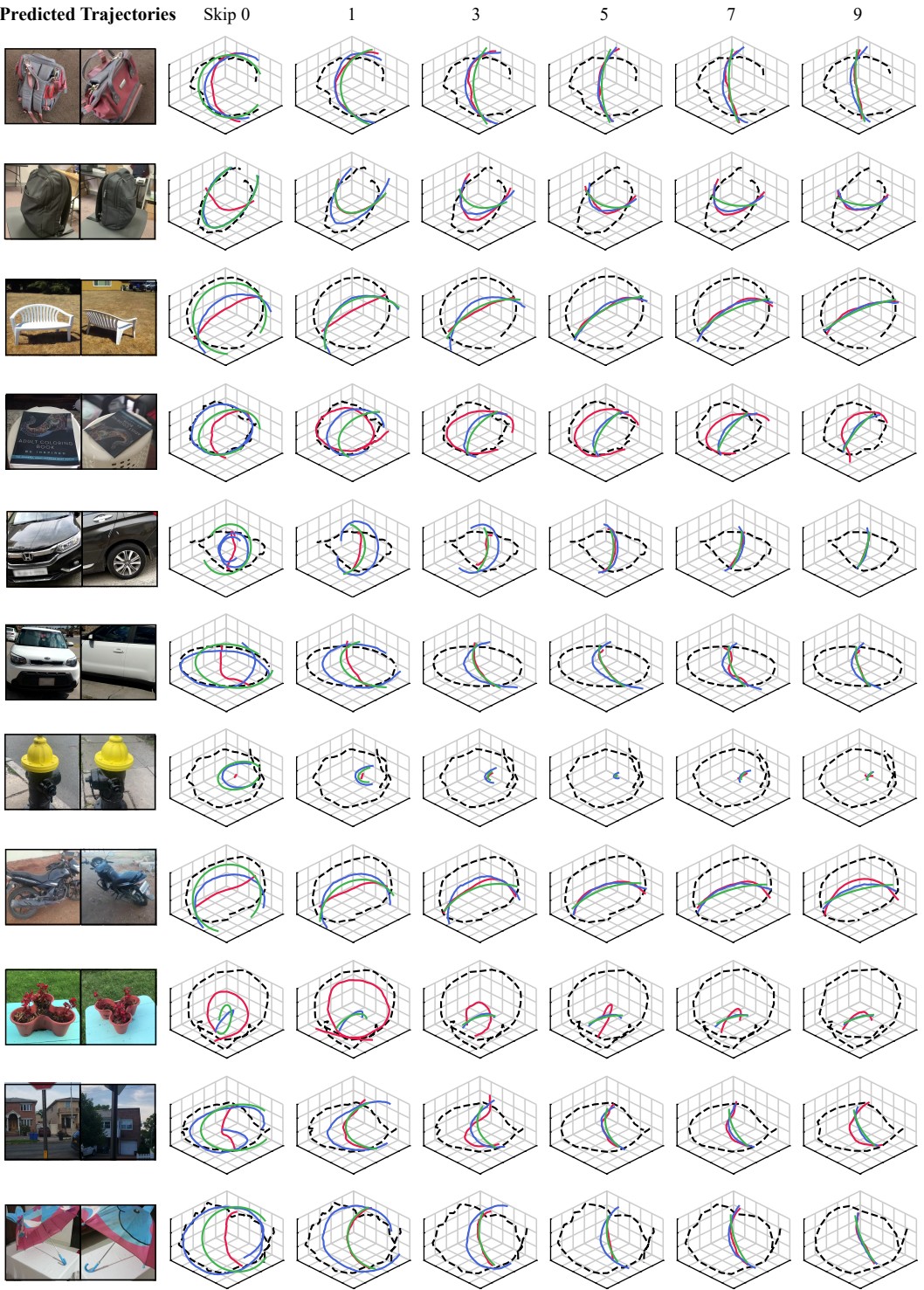

Figure 7: **Failure Cases.** Select failure cases. Interestingly, NIso appears to consistently fail when the background and foreground objects are respectively far from and close to the camera. In such cases, depth maps are often ill-defined, suggesting NIso may encourage models to reason about the underlying 3D structure of the scene.

