# OpenReview forum: "Neural Isometries: Taming Transformations for Equivariant ML"
_NeurIPS.cc/2024/Conference — NeurIPS 2024 poster_

### Official Review · Reviewer_Sfsw · 2024-06-30

**Soundness:** 3
**Presentation:** 3
**Contribution:** 3
**Rating:** 4
**Confidence:** 3

**Summary:**

The paper proposes an autoencoder framework that encodes the input symmetries into isometries in the latent space. The equivariance in the latent space is captured by a functional map $\tau$, which is regularized to be an isometry. Instead of hard constraints, the equivariance of the system is encouraged via the optimized $\tau$ and the loss function design. The proposed method is evaluated on a homography-perturbed MNIST, conformal shape classification, and camera pose estimation tasks. The experiments show the proposed method performs on par with the handcrafted equivariant baselines and outperforms a baseline with a similar approach (Neural Fourier Transform).

I’m positive about the paper. However, there are some main concerns that I would like to know the answers to. I’m willing to raise the scores once they are addressed.

**Strengths:**

1. The paper is well-written and easy to follow.
2. The paper proposes a novel framework that softly models the latent space equivariance as functional maps.
3. Unlike handcrafted equivariant networks that are limited to certain groups, the proposed framework can model different input transformations.
4. The proposed method is shown to perform on par with the baselines in homNIST and camera pose estimation tasks, and outperforms the baselines in the conformal shape classification task.

**Weaknesses:**

1. Although being shown to be effective in the experiments, the constraint of $\tau$ being an isometry seems a bit heuristic. The paper can benefit from some theoretical investigation on why constraining to being isometries helps the performance.
2. Following up on the previous point, can all transformations be modeled as isometric functional maps in the latent space? That being said, does restricting $\tau$ to isometries affect the types of transformations the network is able to model?
3. Since the equivariance is only obtained softly via the loss function design, it’s essential to see how much equivariant is maintained/lost in the latent space. I recommend the authors report some measures of equivariance and compare them with the handcrafted methods. (For example, equivariant error/loss in the latent space.)
4. There are several heuristic designs in the proposed framework, and some of them affect the performance significantly. For example, it is not explained clearly why a smooth multiplicative mask is required to ensure $\tau_\Omega$ is semi-diagonal. The effect of such approximation is also not discussed/evaluated. The second is the multiplicity loss, which greatly affects the performance of the framework.
5. I appreciate the comparisons with handcrafted equivariant networks. However, I believe the paper can be greatly strengthened by comparing it with the same autoencoder framework while not enforcing the latent equivariance and training with data augmentation. This can demonstrate the benefits of the proposed latent equivariant design.

**Questions:**

1. Since there are no hard-coded constraints (except for the regularization) in the system, the proposed method can work with different input transformations. Do you think this work can be applied to automatic symmetry discovery tasks?
2. I understand that the inner product needs to be preserved to construct an isometry, but why does $\tau$ have to commute with $\Omega$?
3. Ln 207, “we show in experiments that a major benefit of our isometric regularization is that our multiplicity loss (promoting a diagonal-as-possible and thus sparse and compact $\tau_\Omega$) can serve as an effective substitute for access to triples.” Is this true for all three experiments, or is it only validated on homNIST?

**Limitations:**

As mentioned in the paper, the proposed network is unable to perform graph-based tasks. In addition, the equivariance in the latent space is not evaluated. As a result, it is unclear how the equivariance is preserved in the latent space. Lastly, by restricting the functional maps to be isometries, the transformations that the network can model might be limited.

---

> ### Author Rebuttal · Authors · 2024-08-05
>
> We thank the reviewer for highlighting that our paper is well-written and easy to follow, novel, and competitive with baselines, and for providing valuable feedback! We further appreciate your commitment to reconsidering your score - we have tried our best to address your feedback, but please let us know if we missed anything!
>
> ## Clarification: Automatic Symmetry Discovery
> We would like to clarify that this is *exactly* the goal of NIso. It *does not assume* upfront knowledge of the symmetry, but instead learns to be equivariant *automatically*.
>
> ## Why commute with $\Omega$?
>
> The goal of NIso is to discover symmetries (i.e. what is preserved between observations) of transformations. For instance, if the image transformation is a shift, then a a particular property of the images is preserved: their frequencies.  For any transformation that preserves some property, there exists a basis such that its functional map (FM) is block-diagonal in that basis. The basis is defined as the eigendecomposition of some operator $\Omega$. The FM being block diagonal in the basis – with the size of each block corresponding to the eigenvalue multiplicity –  means it will commute with $\Omega$ and equivalently preserve its frequencies (i.e. an isometry).  Hence, we can view the problem of symmetry discovery as equivalent to jointly finding a set of maps relating observations *as well as* the operator they commute with.
>
> We will be happy to include a detailed theoretical exposition of these properties in the revision.
>
> ## Can all transformations be modeled?
> NIso can *exactly* model unitary transformations. Any other transformation can in theory be approximated with a sufficiently expressive latent space. We also note that some transformations that are not unitary in image space can be made so by estimating the correct latent space: consider the case of camera pose estimation from video. From frame to frame, there are occlusions so there exists no unitary map between the two images. However, there exists a unitary map between the state of the underlying 3D scenes. Our motivation with NIso is to take a first step towards a model that can discover the correct symmetries of the underlying problem.
>
> Functional maps can nevertheless be sensitive to occlusion and partiality.  However,  we note that NIso is already *surprisingly robust* to partiality in its inputs, as is evident in the Co3D experiments. Please see the related discussion in the general rebuttal.
>
> ## Report measures of equivariance and compare them with the handcrafted methods.
> We are happy to report that we were able to quantitatively evaluate the equivariance of our model following standard procedures [6-7, 43].  Please see the discussion in the general rebuttal and Tab. 1 in the attached PDF.
>
> ## Explanation of why a multiplicative mask is required to ensure diagonality.
>
> The multiplicative mask $P_\Lambda$ appears in the derivation of the closed form solution to the constrained minimization problem in Equation (7) (section A.2 of the supplement).  Remarkably, the derivation reveals that enforcing commutativity with the diagonal matrix of eigenvalues $\Lambda$ (and thus block diagonality) simply reduces to element-wise multiplication with the “sharp” eigenvalue mask in Equation (19) before the Procrustes projection.
>
> ## Evaluate effect of smooth multiplicative mask.
> The “sharp” eigenvalue mask in Equation (19) is difficult to work with.  Evaluating equality between floating point values is hard without some heuristic measure of closeness. Furthermore, we found that this approach results in poor backwards gradient flow to the eigenvalues, to the point where the NIso fails completely.
>
> Thus, we replace the “exact” mask with its “soft” analogue, approximating the Kronecker deltas via the exponential. While it induces some degree of error, it allows backprop through the mask. We note that we are not the first to introduce a “soft” eigenvalue mask with alternatives explored extensively in [Ren and Panine et al. 2019] and concurrent work [Ceng and Deng et al. 2024] employing a similar soft mask as a regularizer.
>
> ## Explanation of Multiplicity Loss.
> The point of the multiplicity loss is to make the operator “interesting” and force our framework to recover the symmetry of the transformation. Without the multiplicity loss, all eigenvalues could collapse to a single one. The operator is then  a multiple of the identity, which would not constrain the FM in a meaningful way. The more distinct eigenvalues the operator has, the more it is forced to discover subspaces of the latent space that are preserved, i.e., symmetries. Intuitively, the multiplicity loss can also be seen as forcing the network to discover an "as-simple-as-possible" representation for the transformations by promoting diagonality.
>
>
> ## Effect of multiplicity loss across experiments
>
> Without the multiplicity loss our method achieves poor results in all three experiments. On Conf. SHREC`11, without the multiplicity loss NIso fails to discover a meaningful notion of equivariance in the latent space and archives a classification accuracy of only approximately 36%.  In the CO3D experiments, we also indirectly evaluate its absence in the regime where we train and evaluate a version of NIso that does not learn an operator (and thus only enforces orthogonality). We find that this version performs worse than all other baselines. Together these results suggest that forcing the network to discover an operator with a diverse eigenspectrum via the multiplicity loss is integral to gaining a meaningful notion of equivariance in the practical sense.
>
>
> ## Compare to the baseline autoencoder framework with data augmentation.
>
> Thank you for suggesting this experiment and we are happy to report we executed it with strong results -- NIso still outperforms these baselines by a significant margin. Please see the "Comparison with data-augmentation" section in the general rebuttal and Table 2 of the PDF.

---

> ### Author Response · Authors · 2024-08-11
> **Address remaining questions in discussion period?**
>
> Dear Reviewer Sfsw,
>
> Thank you again for your comments and feedback! As the end of the author / reviewer discussion period is fast approaching, we would love to hear your thoughts and see if we can address any remaining questions in the remaining time. Please let us know whether we addressed your comments and questions appropriately!
>
> Thank you!
>
> Best,
> the authors

---

> ### Comment · Reviewer_Sfsw · 2024-08-11
>
> I thank the authors for their efforts in the rebuttal. Many of my concerns have been addressed. It's good to see NIso can achieve a similar equivariance error to the handcrafted methods. It's also interesting to see patterns similar to spherical harmonies appear in the SO(3) case. It is also great to see NIso perform better than AE with augmentation. The added experiments strengthen the paper.
>
> In addition, I kindly disagree with other reviewers on the need to solve occlusion/partiality. Occlusion and imperfect symmetry modeling can be a different research topic on its own in the field. However, I recommend that the authors refrain from claiming that NIso can generally handle occlusion/partiality robustly, as it requires additional proof and verification that is not presented in the paper.
>
> Although most of my concerns have been addressed, there are still some that remain. The multiplicity loss still seems like a heuristic trick to me, and it has a significant impact on the system's performance. I would love to see a more theoretical probe into it in future work. Secondly, although it is shown experimentally that the proposed method can deal with transformations that are not unitary, it is still not guaranteed theoretically. The authors provide a plausible explanation in response to Reviewer FFcP (there exists a domain in which these transformations are either unitary or isometric, and can be mapped to via a sufficiently expressive autoencoder). Although I personally agree with such an explanation, this is still a speculation without proper proof. I again suggest the authors refrain from making such a claim.
>
> Overall, the paper is strengthened after the rebuttal, and I believe the paper introduces a novel way to softly model equivariance in the latent space, which is a great addition to the community. Therefore, I have raised my score to reflect the above points.

---

> ### Author Response · Authors · 2024-08-12
>
> Thank you for your response and for updating your score, we are glad that you find NIso will contribute to the NeurIPS community!
>
> Following your comments, we will update our paper to make sure to not overclaim. Specifically, we will clarify that (1) NIso can only exactly model unitary transformations; and (2) That our experimental results only suggest the potential of NIso to model more complex, non-unitary transformations, and that a rigorous theoretical investigation combined with extensive empirical results is necessary to convincingly demonstrate broad generalization. We will also clarify that (3) NIso does currently explicitly handle partiality.
>
> We agree that a deeper investigation into the multiplicity loss will strengthen our method and we are actively investigating this in follow up work. We would also like to note that an important novel benefit of our formulation is that it allows for us to regularize for block-diagonality *without backpropagating these gradients through the solve for the estimated transformation.*  Specifically, the prior methods including the NFT attempt to impose a block-diagonality loss on the estimated transformation itself. We discussed with the authors who confirmed that this is only possible in a post-processing step as it otherwise destabilizes training. In contrast, by learning to parameterize an operator $\Omega$, we instead impose the regularization for block diagonality on the mask $P_\Lambda$ which is stable during training. Our comparisons with the NFT (see Tab. 3, center left) suggest that this is a key feature enabling self-supervised discovery of sparse and condensed representations, and we are excited to investigate this further.
>
> Thank you again for your comments, which will help to clarify the paper!

---

### Official Review · Reviewer_FFcP · 2024-07-08

**Soundness:** 3
**Presentation:** 3
**Contribution:** 2
**Rating:** 4
**Confidence:** 2

**Summary:**

This paper introduces Neural Isometries, which is an autoencoder framework learning to map the observation space to a general-purpose latent space wherein encodings are related by isometries whenever their corresponding observations are geometrically related in world space. Several experiments, including camera pose estimation, are conducted.

**Strengths:**

The idea of transforming complicated equivariances in observation space into isometries in latent space is intuitive and interesting. Experiments have demonstrated the potentials of this method in practical tasks. The paper is well written and easy to understand.

**Weaknesses:**

This is an interesting paper but in-depth discussion is lacking. My main concern is that there is no quantitative or theoretical description of the equivariance. For example,
1.  How equivariant is the model? i.e. How large is the equivariance loss?
2. What type of equivariance in the observation space can be modeled? Will the model fail when the pertubations are too strong?

In addition, the experiment results are relatively weak. Only 3 small scale experiments are conducted, and only a few baselines are considered.

**Questions:**

1. If the observation space is already isometric, e.g. rotate by 90 degrees, what can be said about the latent sapce?
2. How robust is this model? For example, how will the latent vector change when the inputs are partially observed (masked)?

**Limitations:**

See the weakness section.

---

> ### Author Rebuttal · Authors · 2024-08-05
>
> Thank you for validating our motivation and our communication - we appreciate it!
>
> ## Quantitative evaluation of equivariance: How equivariant is the model?
>
> We are happy to report that we were able to quantitatively evaluate the equivariance of our model following standard procedures [6-7, 43].  Please see the discussion in the general rebuttal and Tab. 1 in the attached PDF.
>
> ## What type of equivariance in the observation space can be modeled?
>
> NIso can *exactly* model unitary transformations. Any other transformation can in theory be approximated with a sufficiently expressive latent space. We also note that some transformations that are not unitary in image space can be made unitary by estimating the correct latent space: consider the case of camera pose estimation from video. From frame to frame, there are occlusions so there exists no unitary map between the two images. However, there exists a unitary map between the state of the underlying 3D scenes. Our motivation with NIso is to take a first step towards a model that can discover the correct symmetry of the underlying problem.
>
> ## Will the model fail when the pertubations are too strong? How robust is the model?
>
> As remarked in the general discussion, functional maps can be sensitive to occlusion and partiality.  However, we note that NIso is already *surprisingly robust* to partiality in its inputs, as is evident in the Co3D experiments. Please see the discussion in the general rebuttal under the “NIso under the presence of occlusions / masking / partiality” section.
>
>
> ## Lack of Baselines + Only Small-Scale / Synthetic Experimental Results
>
> We are happy to report that we have provided comparisons with additional baselines, including with two SoTA representation learners (DINOv2 and BeIT) in the pose estimation experiments on the Co3D dataset. Please see the "Additional baselines" section in the general rebuttal.
>
> However, we would like to respectfully push back on the notion that we are considering too few baselines.
>
> We note that there is not much work on the topic of equivariant machine learning *without* prior knowledge of the symmetry group - the NFT is the most relevant baseline in this space. Further, the reviewer is suggesting that we are only comparing “small-scale” or “synthetic” experiments”. However, camera pose estimation on Co3D is *not* a small-scale experiment - it is a large, real-world dataset that is actively used for benchmarking of applications from pose estimation to novel view synthesis.
>
> More generally,  we sought geometric deep-learning baselines for difficult, non-compact symmetry groups. To the best of our knowledge, the only existing SoTA equivariant models handling such symmetries in vision-related tasks are are homConv [6] and LieDecomp [43] (both for homographies) and MobiusConv [7] (for Mobius Transformations). Thus, benchmarking on MNIST and SHREC is absolutely vital to support our claims. These baselines are not scalable and will not run on any more complex or real-world datasets - this is not a limitation of *our* method, but a limitation of the baselines.  Please also see our discussion in the general rebuttal under the section "Scope of experimental evaluations".
>
> We would be happy to compare with additional baselines if you could clarify what these baselines should be.
>
>
> ## If the observation space is already isometric, e.g. rotate by 90 degrees, what can be said about the latent space?
>
> Note that our toy experiment - discovering the toric laplacian - is *exactly* an instance of this problem. We parameterize images to lie on a torus, such that shifts become rotations, and hence, the shift is exactly an isometric transformation! In this case, our formulation is likely to recover a close approximation of the *correct* operator and corresponding irreducible unitary representation of the transformation in the eigenbasis - in Exp. 1, we recover a very good approximation of the laplacian operator and the block-diagonal shift representation.
>
> We further demonstrate this in the new experiment where we discover the spherical harmonic transform (see the “Discovering the spherical harmonic transform” section in the general rebuttal and Fig. 1 of the attached PDF).  This experiment results in the discovery of a basis with almost *exactly* the properties of the spherical harmonics. In particular, the estimated $\tau_\Omega$ manifest *exactly* the same structure as the ground truth Wigner-D matrices corresponding to the rotation (which are the IURs of $SO(3)$), with square blocks of size $(2\ell + 1) \times (2\ell + 1)$, for the $\ell$-th distinct eigenvalue

---

> > ### Comment · Reviewer_FFcP · 2024-08-09
> >
> > Thanks for your clarification. I have read the experiments in the attached pdf, and I am clear now that the equivariance error is at least comparable with the existing methods. The reply on the experiments is also convincing.
> >
> > However, the other two concerns are not addressed: the type of equivariance can be modeled and the robustness to partial visibility.  The author argued that  "Any other transformation can in theory be approximated with a sufficiently expressive latent space". It is imprecise.
> >
> > On the other hand, due to the lack of description of the robustness against partial visibility, and the model does not have any constraint on the equivariance of the input, it is hard to understand why the model is robust or how robust I should expect it to be for practical tasks. The author mentioned that this issue will be investigated in the next step, but since current version of paper does not include any discussion on this issue, I think it is not complete.

---

> ### Author Response · Authors · 2024-08-10
> **Response (1/2): Modeling Equivariance**
>
> Thank you for your response. We are glad to hear that you view our experimental results as convincing.
>
> ## Which Types of Equivariance can be modeled?
>
> We apologize for the lack of clarity in our previous response. In the following, we will make two points: (1) We show that NIso can exactly model equivariance for any unitary transformations, which make up a large chunk of previously proposed equivariant ML architectures. (2) We provide detailed reasoning for our empirical results that clearly suggest that NIso can learn to be equivariant even to certain non-unitary groups.
>
> ### Equivariance to unitary transformations
> NIso can learn to be equivariant to any transformation that preserves the norm of its input. Examples for this include:
> Shifts on the torus, as demonstrated in Fig. 3 of the main paper.
> SO(3) with spherical inputs, as demonstrated in Fig. 1 of the rebuttal PDF.
> The group of 90 degree rotations mentioned by the reviewer, which we empirically validated but were not able to include in the rebuttalPDF due to space constraints (we will include this in the camera-ready paper!)
>
> We note that prior work in equivariant ML at top conferences is often equivariant to only **a single one** of these transformations, with significant expert crafting. For instance, SO(3) is a non-abelian group and its IURs are substantially complex – nevertheless we are able to recover a basis and representation of transformations with  *exactly* the same properties.
>
> The fact that NIso learns to be equivariant to all these transformations is a significant strength - we are not aware of any previous work in symmetry discovery that has demonstrated this capacity.
>
> We hope that this is absolutely un-ambiguous: **NIso can learn to be equivariant to a transformation that preserves a norm of its input**.
>
> ### Equivariance to non-unitary transformations
> We agree that the statement “Any transformation can be approximated with a sufficiently strong encoder” was too loose. We specify it in the following.
>
> As discussed above, NIso can closely approximate **unitary** transformations on its input.
> For other more complex transformations, our experiments suggest that a sufficiently expressive autoencoder can, in fact, discover a latent space where the manifestation of the transformation is approximately unitary and even isometric.  We believe this is a surprising and interesting result that is perhaps being overlooked. We note that it is not understood which transformations or groups can be embedded in this way, and point out that this is an exploration of fundamental math, beyond the scope of any ML paper.
>
> However,  a potential explanation of this phenomenon could be related to the fact that for many challenging transformations, there exists a domain in which these transformations are either unitary or isometric. For example, Mobius transformations are isomorphic to the isometries of the hyperbolic ball with their action on the sphere representing their restriction to the boundary. Similarly, camera motions in the image plane are in fact projections of the isometric action of SE(3) on the underlying 3D scene.
>
> We hypothesize that by forcing the network to find an isometric representation for the observed transformations we are implicitly regularizing the learned latent space to discover information about this underlying geometry. We note that our ability to recover reasonable camera poses from the estimated transformations where other representation learners fail is indicative of this.
>
> An additional illustrative example supporting this hypothesis can in fact be seen in Figure 7 of the supplement, where we show examples of failure cases in which the pose extraction from $\tau_{\Omega}$ fails catastrophically. Here we observe that these failure cases often coincide with sequences of frames where background and foreground objects are respectively far from and close to the camera – both cases where depth-estimation pipelines also often fail.
>
> ### Conclusion
> All in all,  NIso is a novel *first step* towards a new paradigm for symmetry discovery and equivariant ML. To the best of our knowledge, no prior work in symmetry discovery has been able to show, for instance, self-supervised discovery of approximate equivariance to a variety complex transformations, some of which do not even form a group in the observation space (such as the action of camera motions on the image plane).
>
> Using the additional page of the paper, we will add above discussion to the “Methods” section of the paper - we agree with the reviewer that this will make the paper stronger!

---

> ### Author Response · Authors · 2024-08-10
> **Response (2/2): Analysis of Partiality Robustness**
>
> ## Analysis: Robustness to Partiality
> In response to the reviewer’s concerns, we are happy to report that we performed a theoretically grounded experimental evaluation of robustness which we will be happy to include in the revision using the additional page of the camera-ready paper.
>
> Specifically, we seek to evaluate the robustness of the isometry-estimation module in isolation in the presence of occlusions. We consider  the encoder-free paradigm – consisting of the projection to the learned basis, the estimation of the isometric map as in Equation (8), followed by unprojection – which is the same setup in the toric and spherical laplacian experiments. **This setup allows us to study exactly the effect of masking on the latents (as requested by the reviewer) which in this case are the input images.** To do so, we consider two observations $\psi$ and $T\psi$ which only partially correspond and denote  $O_\psi$ and $O_{T\psi}$ to be the diagonal overlap masks such that the $i$-th diagonal element of $O_{\psi}$ is 1 if $i$ is the overlap with $T\psi$ and $0$ otherwise, with $O_{T\psi}$ defined in the same manner. We observe that the equivariance error under the partiality can thus be defined by the magnitude of the difference between the components of $O_{\psi} \psi$ that get mapped to $ O_{T\psi} T\psi $ under $\tau$. That is, we define the partiality equivariance error to be $  \lVert  O_{T\psi} \tau O_{\psi} \psi - O_{T\psi} T\psi \rVert^2 / \lVert O_{T\psi} T\psi \rVert^2$.
>
> Using the toric laplacian experiments as a base, we consider two models of partiality. In the first, we no longer consider the domain to be toric and instead shifted images are clipped at the boundaries with the resulting empty pixels masked to zero – corresponding to the type of partiality often observed in video. In the second, we randomly mask out $2 \times 2$ patches in the shifted image. We train five instances of our model under both partiality regimes, masking out approximately 10%, 20%, 30%, 40%, and 50% of pixels in each instance and measuring the resulting partial equivariance error on the test set. The results are shown in the table below.
>
> | Percent Occluded  | 10% | 20% | 30%| 40%| 50% |
> | :------------------------ | :-----: | :-----: | :-----: | :-----: | :-----: |
> | Shift Mask    	|   5.66%   | 9.42% | 14.43% | 17.42% | 20.47% |
> | Patch Mask      	|   6.69%  | 13.62% | 23.54% | 34.14% | 45.35%|
>
> Notably, we see that the partial equivariance error is consistently lower in the presence of shift-based occlusions, and  increases less as the percentage of occluded pixels increases. We observe that the principal difference between these two regimes is that the unoccluded pixels exist in a contiguous block under the shift mask, whereas the region of unoccluded pixels is fragmented under the patch masking. Intuitively, the contiguous matches between large blocks (comprising the majority of the image) act as a strong regularizer that de-prioritizes matching the occluded areas. Conversely, when the correspondence is interrupted and fragmented it offers weaker regularization and the resulting map is more affected by the spurious matches induced by the occlusion.

---

> ### Author Response · Authors · 2024-08-12
> **Additional question remaining in discussion?**
>
> Dear FFcP,
>
> Thank you again for your response and feedback! We believe we were able to address your stated remaining concerns, including a straightforward experimental evaluation of robustness.  As the end of the author / reviewer discussion period is fast approaching, we would like to be sure you are satisfied by our discuss and see if we can address any remaining questions. Please let us know whether we addressed your comments and questions appropriately!
>
> Thank you!
>
> Best, the authors

---

### Official Review · Reviewer_JQqD · 2024-07-14

**Soundness:** 2
**Presentation:** 4
**Contribution:** 3
**Rating:** 6
**Confidence:** 3

**Summary:**

This paper proposes a generic equivariant ML framework by learning latent representations are are modelled to be related by an isometry. There are several important design choices made by the authors: (1.) An autoencoder framework that keeps the spatial structure (i.e. images get encoded into images) (2.) The isometric map is represented compactly with a basis that is the outcome of an eigen- -decomposition of a PSD operator (3.) The isometry manifests as an orthogonal matrix with a sparse block diagonal structure in its reduced projected form.

Experiments are shown for (1.) Learning the spectrum of a Toric Laplacian and comparing it qualitatively to the real one  in Fig 3 (2.) Homography perturbed MNIST (3.) Conformal shape classification and (4.) Camera Pose Estimation from real-world video. The results (1) show a good proof of concept and (2), (3) and (4) demonstrate a clear superiority over the closest conceptual baseline NFT. (2.) Also makes the case for achieving comparable performance with simpler equivariant architectures using the NIso latent space.

**Strengths:**

- Overall the biggest strength of this paper is that it was an enjoyable read. I found the hypothesis (i.e. learning latent codes related by an isometry for arbitrary transformations in the observation space) to be interesting and reasonable. The overall writing and description of the method is very nice.
- The Experiments especially (1.) was a good proof of concept.

**Weaknesses:**

- I am actually quite confused on how the operator \Omega and Mass matrix M is learned? The Laplace Beltrami is a very structured object. See all requirements enumerated in Wardetzky, Max, et al. "Discrete Laplace operators: no free lunch." Symposium on Geometry processing. Vol. 33. 2007.). It is unclear what the nature of this operator is and its learning feels ad-hoc and unmotivated.
- Do the authors make sure that the optimization for the functional map is well-posed? Typically in the original formulation a "good" functional map is the outcome of using many-many descriptors in addition to regularization constraints like commutativity with laplacian. If I understand correctly the proposed formulation seems to be confident in learning a structured functional map (like Fig 2) using just one pair of corresponding functions ?
- As a high-level opinion: Overall the experimental section could be stronger. The "discovery" of the toric laplacian was nice, but a demonstration on roto-translations perhaps with scale and/or spherical images with SO(3) would be extremely convincing. I would also recommend an evaluation that focuses beyond NFT and actually quantifying the complexity of different equivariant models.
- As mentioned by the authors, the requirement of generalizing to domains with diverse connectivity is indeed a hard one, but important.

**Questions:**

- The basis functions visualized in Fig3 (third column) are learned or from the actual toric laplacian? what are the various rows?
- It would also be useful to report some feature of complexity in Table 1 indicating "meticulously-engineered, handcrafted networks"
- More examples of evidence showing good basis functions
- What is a perturbation in Section 5.1?

**Limitations:**

I think the authors make a fair declaration of the limitations of their work in Section 6.
I would make a stronger vote after the rebuttal. I think this paper has many ingredients to warrant acceptance: clever idea, good argumentation, and decent proof of concept. It lacks clarity in some places and the evaluation does not appear equally general as is the description of the method. For this reason, I maintain a borderline acceptance with a full intention to re-assess in the next round.

---

> ### Author Rebuttal · Authors · 2024-08-05
>
> Thank you for your detailed review, and for your kind words on our hypothesis being interesting and plausible, our writing enjoyable, and our validation with the laplacian helpful! We are happy to report that we executed your key task -- recovering the harmonics with spherical images – with exciting results!
>
> ## Learning the operator $\Omega$ and mass matrix $\mathbb{M}$.
>
> ### Motivation
>
> The goal of NIso is to discover symmetries (i.e. what is preserved between observations) of complex transformations in image space. For instance, in the toric laplacian experiment, the image transformation is a shift. Under shifts, a particular property of the images is preserved: their frequencies.  For any transformation that preserves some property, there exists a basis such that its functional map (FM) is block-diagonal in that basis. The functional basis is defined as the eigendecomposition of some operator $\Omega$. The FM being block diagonal in the basis – with the size of each block corresponding to the eigenvalue multiplicity –  means it will commute with $\Omega$ and equivalently preserve its frequencies (i.e. an isometry).  Hence, we can view the problem of symmetry discovery as equivalent to jointly finding a set of maps relating observations *as well as* the operator they commute with.
>
> ### Parameterization and learning
>
> [Wang and Solomon, 2019] establish that the key property characterizing functional operators is semi-positive-definiteness (SPD), and we impose no other constraints so as not to limit the space of discoverable operators. (We will cite this work in the revision). SPD is defined with respect to some  inner product, and thus we learn a diagonal mass matrix (with positive entries) which defines a functional inner product. Then, we seek to simultaneously regress the parameters for an operator $\Omega$ that is SPD w.r.t. the inner product defined by $\mathbb{M}$. We observe that *any* $\mathbb{M}$-SPD operator can be expressed in the form of Equation (4) for some matrix of $\mathbb{M}$-orthogonal eigenfunctions $\Phi$ and non-negative eigenvalues $\Lambda$. Thus we learn a set of weights parameterizing $\Phi$ and $\Lambda$ which together with learned $\mathbb{M}$ form $\Omega$.  In practice, the weights of  $\Phi$ are projected to the nearest $\mathbb{M}$-orthogonal matrix via the SVD.
>
> ## Well-posedness of functional map
> We carefully ensure that the solve for the functional map is well-posed.
>
> First, we leverage a number of regularizers that jointly ensure well-posedness: 1). Orthogonality; 2). Operator commutativity;  and 3). The “Equivariance loss” is a version of the widely used “descriptor preservation” loss in the FM. These are several of the most common principled regularizers widely disseminated and discussed in the FM literature [21, 26, 38]
>
> Second, we note that our latent codes double as multi-channel pointwise descriptors, of the same kind as used in the Deep FM family of work [Litany et al. 2017, 38] , which forms the backbone of many SoTA shape correspondence pipelines.
>
> ## Discovering the Spherical Harmonic Transform
> We are happy to report that we have performed this experiment with *spectacular* results! Please see the discussion in the general rebuttal under “Discovering the spherical harmonics” and Fig. 1 of the PDF. We hope you find these convincing!
>
> ## Quantifying the complexity of different equivariant models.
> There are two dimensions of “complexity” that we failed to clearly delineate in the text: (1) *human* complexity of hand-crafting the method and (2) compute / memory complexity.
>
> While approaches for rotation equivariance are wide-spread, there are no standard equivariant networks for most difficult symmetries. Any attempt at building such a network is associated with extensive, difficult mathematical labor. MobiusConv [7], for instance, essentially derives what to the best of our knowledge is a previously unknown representation on a specialized group of filters (which is no mean feat), simply to facilitate a tractable discretization.
>
> homConv [6] and LieDecomp [43] seek to sidestep this via a recipe for general equivariance using Monte-carlo integration. However, these methods struggle with *compute* complexity. For $N$ samples (typically at least 10-25), the computational complexity of a forward pass scales as $N^L$ where $L$ is the number of layers!  In practice, this means that these methods cannot be scaled past incredibly low-dimensional images, which is why both homConv and LieDecomp are run only on MNIST.
>
> In contrast, NIso relies on recovering a latent space such that the transformations can be represented as an isometry, which can thus be exploited by far more efficient and scalable isometry-equivariant networks such as [36] and [42]. .
>
> ## Evaluation that focuses beyond NFT.
> We agree and have since added two additional baselines to the submission - see the point “Additional Baselines” in the general reply above! However, we would like to note that besides the NFT, *we are not aware of any other methods that aim to perform equivariant representation learning without prior knowledge of the group*. Thus, we see the NFT as a important benchmark with which to compare.
>
> ## Generalizing across connectivity
> We completely agree and are currently exploring this as a direction of future work.
>
> ## The basis functions visualized in Fig3 (third column) are learned or from the actual toric laplacian? What are the various rows?
> They are learned. At the time we did not discover them ordered by a classical notion of frequency so the order is random.  Rows are eigenfunctions in C-style indexing.
>
> ## More examples of evidence showing good basis functions
> In addition to the learned spherical harmonics, we have also visualized examples of the eigenfunctions discovered in each of the three other experiments.  Please find them in the PDF, in Fig. 2.
>
> ## What is a perturbation in Section 5.1?
> Perturbation refers to applying a homography to the image.

---

> ### Author Response · Authors · 2024-08-11
> **Address remaining questions in discussion?**
>
> Dear JQqD,
>
> Thank you again for your comments and feedback! As the end of the author / reviewer discussion period is fast approaching, we would love to hear your thoughts and see if we can address any remaining questions in the remaining time. Please let us know whether we addressed your comments and questions appropriately!
>
> Thank you!
>
> Best,
> the authors

---

> > ### Comment · Reviewer_JQqD · 2024-08-12
> > **Rebuttal response**
> >
> > Thank you for a refreshing rebuttal. After reading through the rebuttal and other reviews, I am quite happy to raise my score to a clear acceptance. This is an interesting paper with some fascinating observations. However, there is still some way to go before this can really be generalized and put into meaningful action. Sustaining drawbacks are some very hard questions like - how much data is needed for this approach? Equivariance to multiple non-trivial (perhaps non-unitary) transformations of the input etc. But I think as a good proof of concept, I believe this paper tells a story that would be interesting at Neurips. I would highly recommend including the spherical image experiment in the main draft (also what kind of architecture did you use for the encoder?)

---

> > > ### Author Response · Authors · 2024-08-12
> > >
> > > Thank you for your response and we are extremely glad to hear you are happy to recommend a clear acceptance!
> > >
> > > We completely agree that NIso represents a first step towards identifying underlying symmetries and that further investigation is necessary to support claims of broad generalization. The reviewer is correct to point out that answering difficult questions like those regarding data efficiency and effectiveness in the presence of multiple, different non-unitary actions are important prerequisites. We will amend the revision to reflect this and make sure that we communicate limitations on partiality and transformations modeled clearly.
> > >
> > > We will incorporate the spherical harmonic transform experiment into the main body of the revision as the results are very compelling (thank you again for suggesting this experiment!). In both this experiment and the toric laplacian experiment, the encoder and decoder are just the identity map. We pass the images directly to the transformation estimation module, learning only the parameterization of the operator with the goal of evaluating the abilities of the module in isolation.

---

### Official Review · Reviewer_2Npe · 2024-07-15

**Soundness:** 2
**Presentation:** 2
**Contribution:** 2
**Rating:** 4
**Confidence:** 4

**Summary:**

The main motivation behind Neural Isometry is as follows: most of real world transformations in vision and geometry processing lack identifiable group structure and therefore challenging for prior work in equivariant learning that assumes such knowledge apriori. The paper proposes an auto encoder framework that maps observations (related by some geometric transformations) to a latent space where embeddings are  related by a linear transform. The framework leverages the existing functional map framework and corresponding regularization to learn this latent space. The paper validates and compares their performance with the baseline NFT (Neural Fourier Transform) that assumes the knowledge of the group a priori.

**Strengths:**

-  The problem of learning structured latent space in a self supervised equivariant learning set up is relevant.

- The proposed framework is formalized clearly and in detail in Section 4.

**Weaknesses:**

As mentioned many times in paper, Instead of theoretical justification ( as in NFT), this paper seeks to validate the efficacy of approach experimentally on real world tasks.

- Limited Experimental Set up: The datasets used are still very small scale (MNIST,) or synthetic (SHREC) and therefore far from the real world data.  The submission mainly follows and compares with one baseline (NFT) that was implemented by authors themselves.  The submission could motivate the problem/solution by probing/playing with the learned structured space e.g. in geometry processing tasks [1].

- Limitation of Functional map: One of the biggest limitation of functional map framework is its applicability to real world data where partiality is ubiquitous (due to occlusion etc). Therefore, Neural Isometry by design also inherits this limitation.  experiments are shown on MNIST/SHREC where there is no occlusion or partiality. In the third benchmark, instead of skipping the odd frame, does the performance decays rapidly between distant frames?

- Presentation: The submission contains several typos, convoluted sentences (Line 85-88) and statements without a citation. e.g.

-- Line 49: < Lorentz transformations with d'almbert operator of Minkowski space> citation is missing here. Also, why do we need to know this fact in the introduction of this paper?

-- Line 42: <by preserving the spatial dimensions> not sure what does this mean after reading the next 2 lines. Please clarify

-- Line 89: <orthogonal relaxation of FM> citation or please detail what those are.

- Related work: The submission cites a dozen variants of deep functional map paper [2] but not the deep functional map paper itself [2] that inspired them. Please justify how [20,21,22,39] are related to this work or inspired this work given other deep functional pipeline cited in paper. The reviewer would instead relate/distinguish this work with [3].


1. Composite Shape Modeling via Latent Space Factorization, Dubrovina et al, 2021
2. Deep Functional Maps Litany et al. 2017
3. Map-based exploration of intrinsic shape differences and variability  Rustamov et al. 2013

**Questions:**

please see above.

**Limitations:**

The submission should state the scalability of this approach given its reliance on Laplacian Eigen basis.

---

> ### Author Rebuttal · Authors · 2024-08-05
>
> ## Limited experimental setup
> We would like to respectfully push back on the notion that we are considering too few baselines. We note that there is very little work on equivariant machine learning *without* prior knowledge of the symmetry group - the NFT is the most relevant baseline in this space. Further, the reviewer is suggesting that we are only comparing “small-scale” or “synthetic” experiments”. However, camera pose estimation on Co3D is *not* - it is a large, real-world dataset that is actively used for benchmarking of applications from pose estimation to novel view synthesis.
>
> More generally,  we sought geometric deep-learning baselines for difficult, non-compact symmetries in vision tasks. To the best of our knowledge, the only existing models handling these symmetries are homConv [6] and LieDecomp [43] (both for homographies) and MobiusConv [7] (for Mobius Transformations). Thus, benchmarking on MNIST and SHREC is necessary to support our claims. These baselines are not scalable and will not run on any more complex or real-world datasets - this is not a limitation of *our* method, but a limitation of the baselines.
>
> We would be happy to compare with additional baselines if you could clarify what they should be.
>
> ## Limitation of functional maps (partiality)
> We agree that functional maps can be sensitive to occlusion and partiality.  However,  we note that NIso is already *surprisingly robust* to partiality in its inputs, as is evident in the Co3D experiments. Please see the discussion in the general rebuttal.
>
> ## Presentation
>
> ### Line 49: < Lorentz transformations with d'almbert operator of Minkowski space...
>
> Thank you - we agree that this is confusing and removed it.
>
> ### Line 42: <by preserving the spatial dimensions...
> By this, we mean that the encoder encodes images into 2D feature maps with height and width higher than 1x1 pixel. This is opposed to *collapsing* the spatial dimension, as done in the NFT, which encodes the whole image into a single, global latent vector.
>
> However, we agree that this is unclear as written, and will change the wording.
>
> ### Line 89: <orthogonal relaxation of FM>
>
> Reference [23] (cited later in the same sentence) seeks functional maps that represent near conformal, rather than isometric deformations. Conformal transformations preserve the Dirichlet inner product, and thus manifest as orthogonal transformations in the eigenbasis of the Dirichlet Laplacian.
>
> That said, we agree that the sentence is overly long and that “orthogonal relaxation” is vague as written. This will be addressed in the revision.
>
> ### Does not cite the “Deep Functional Maps” paper inspiring cited work.
> We will add this in the revision and make clear its influence.
>
> ### Justify how [20, 21, 22, 39] are related to or inspire present work
> These represent recent work that either seeks to address existing drawbacks of FM, including scalability [20] and spatial consistency [21], investigate the properties of deep FMs frameworks [39], or explore composability with other powerful tools including LLMs [22]. All methods achieve state-of-the-art results, and we include them to show that increasing the efficacy and flexibility of FMs is an active topic.
>
> However, we would be happy to instead discuss in more detail those works most closely related to our own, including Litany et al. 2017 and Rustamov et al. 2013.
>
> ### Relate/distinguish with the work of Rustamov et al. 2013
> We thank the reviewer for suggesting we examine this work, as we now understand it provides an interesting comparison we did not previously realize!
>
> This work shows how performing PCA on shape difference operators constructed with FMs forms a kind of “latent space” where codes corresponding to given shapes share a notion of closeness whenever said shapes can be related by specific transformations, including conformal and  authalic (area-preserving). Like NIso, this forms a framework for symmetry discovery by observing the clusterings in the latent space. However, in NIso similar observations lie on the same orbit formed by the transformations $\tau$, rather than being “close” in the Euclidean sense.
>
> Perhaps the biggest difference between NIso and this work is that NIso also recovers a dimensionally-reduced representation of the transformations between observations. In addition,  we demonstrate in experiments that NIso can recover a meaningfully structured latent space for various types of transformations the observation space, whereas this work only reveals meaningful structure as it relates to conformal and authalic transformations.
>
> ## Scalability of the eigenbasis
> We agree that the ultimate scalability of the eigenbasis could be a limitation and we will note this in the revision. We believe the limiting factor is the parameterization of the basis directly via learned weights, which would result in large model sizes for high-res latent spaces. Currently, we find this can be mitigated by encoding to a lower resolution latent space, where the dimensions of the weight matrices are manageable, though this likely comes at the cost of some expressivity due to aliasing.  Currently, we are working on a follow up to address this problem in tandem with partiality by defining an *observation-dependent* eigenbasis via the output of a second encoder inspired by the concurrent work of [Cheng et al. 2024].
>
> However, we note that in other Deep FM frameworks, the principal computational bottleneck is the solve for the functional map. For example, the method proposed in the influential [38] to compute a near isometric functional map requires solving $K$  $K \times K$ linear systems, with $K$ the dimension of the eigenbasis. For NIso, the solve requires only a single $K \times K$ SVD computation. This is due to the closed form solution derived in Equation (8) which, while not mathematically significant, is to the best of our knowledge novel in the functional maps literature.

---

> ### Author Response · Authors · 2024-08-11
> **Questions remaining?**
>
> Dear 2Npe,
>
> Thank you again for your comments and feedback! As the end of the author / reviewer discussion period is fast approaching, we would love to hear your thoughts and see if we can address any remaining questions in the remaining time. Please let us know whether we addressed your comments and questions appropriately!
>
> Thank you!
>
> Best,
> the authors

---

### Author Rebuttal · Authors · 2024-08-05

We thank the reviewers for their careful reading, and detailed and considerate feedback. We are glad that reviewers deem our paper “a clever idea”, “relevant”, “interesting and reasonable”, and “intuitive and interesting”, and the writing to be “very nice”, “easy to follow”, an “enjoyable read”, and a “clear formalization”. We are glad that reviewers recognize the relevance of this first step towards self-supervised symmetry discovery and equivariant ML!

Key outstanding concerns revolve around measuring the equivariance error, clarity, and experimental evaluation. We are glad to present new results and analysis to address these concerns!

## Measuring Equivariance Error
We measure the standard equivariance error in the latent space [6-7, 43], with the results shown in Tab. 1 of the PDF. On both HomMNIST and SHREC, NIso is on par with expert-designed baselines. Similarly for Co3D, we find that NIso achieves low errors. While the error rises with increasing frame skip, it remains under 10%, and NIso remains the best-performing method, demonstrating that NIso remains equivariant even under increasing partiality.

## Discovering the spherical harmonic transform
Following JQqD’s suggestion, we perform an experiment identical to “discovering the toric laplacian”, by mapping ImageNet to the sphere and acting on it via SO(3). We further realized that adding a simple dropout layer which randomly masks out coefficients corresponding to large eigenvalues before the basis unprojection yields eigenfunctions ordered by their energy. This experiment results in the discovery of a basis and maps $\tau_{\Omega}$ with almost *exactly* the properties of the spherical harmonics and the Wigner-D matrices, see the attached PDF, Fig. 1!

## Additional baselines
### Pose estimation
In the camera pose estimation experiment, we added two strong representation-learning baselines. We extract image features from both images using two state-of-the-art vision foundation models -  DINOv2 [Oquab et al. 2023] and BeIT [Bao et al. 2021]. We then pass the tokens into the DUSt3R[47]-style decoder described in section 5.3 to predict the pose. The results are shown in Table 2 of the PDF.  NIso outperforms both significantly.

### Comparison with data augmentation
In line with Sfsw’s comments, we compare NIso to the AE baseline with dataset augmentation during both the pre-training and fine-tuning phases in the SHREC and MNIST experiments. The results are shown in the PDF, Tab. 2. On both MNIST and SHREC, training with augmentation improves the baseline performance by approximately 34% and 7%. NIso still outperforms this baseline significantly.

## Scope of experimental evaluations
2Npe remarks a “limited experimental setup”. We would like to contextualize the scope of our evaluation with that of comparable work at top conferences.

HomConv [CVPR] and LieDecomp [ICRL] benchmark *exclusively* on MNIST. MobiusConv [SIGGRAPH] benchmarks *only* on SHREC to demonstrate advantages of Mobius-equivariance. Our experiments are also commensurate with those in the NFT [NeurIPS, ICLR]. Among all these methods, we alone present results on Co3D, a large-scale, real-world dataset, in contrast to 2Npe's review ("datasets are very small scale, synthetic, not real world.").

Further, 2Npe prefaces their criticism with the statement "As mentioned many times in paper, instead of theoretical justification, this paper seeks to validate the efficacy of approach experimentally on real world tasks." We respectfully point out that this is a misrepresentation of our claims.  In fact, *only once*, on lines 101-102 do we make a similar statement in which we state our intention to "validate the efficacy of
our approach experimentally, including in geometry processing and real-world 3D vision tasks" (referencing the conformal SHREC and Co3D experiments).  At no point do we say our goal is to validate NIso on real-world tasks, only that our experiments *include* a real-world task (Co3D pose estimation).

2Npe also remarks that we could "motivate the problem/solution by probing/playing with the learned structured space". As our chief goal is self-supervised symmetry discovery, we respectfully note that our toric and spherical Laplacian experiments serve *exactly* this end.

In terms of baselines, we benchmarked with the *best-performing* ones for homography and mobius equivariance, and SOTA representation learners for Co3D. To the best of our knowledge, the NFT is the only applicable baseline for equivariant representation learning without prior knowledge of the group. We contacted the authors of the NFT, who confirmed our implementation is faithful.

It is challenging to pick the right evaluation when working on the new problem of self-supervised symmetry discovery / equivariant ML. We are hence grateful for the reviewers' suggestions, which we did our very best to incorporate!

## NIso under the presence of occlusions / masking / partiality
We agree that showing robustness to partiality would make the paper stronger. That said, several recent works have sought to make functional maps robust to partiality [Attaiki et al. 2021, Cheng et al. 2024, Bracha et al. 2024] and we are excited to incorporate these techniques in our next steps! However, we strongly believe that NIso without this improvement remains extremely valuable. First, the limitation of not modeling partiality is not unique to NIso, but to most equivariant networks, including all of NIso’s baselines. Second, we note that NIso is already robust to partiality, as is evident in the Co3D experiments. At a frameskip 9, only about 80% of the pixels are shared across frames, and NIso outperforms two foundation model baselines with the equivariance error remaining below 10%. Third, a major application for NIso is the continuous-time regime, i.e., representation learning from video. In this case, the overlap between consecutive frames is generally large,and NIso remains acutely practical.

---

### Decision · Program_Chairs · 2024-09-25

**Decision:**

Accept (poster)

**Comment:**

This paper received mixed reviews in the first round, was intensively discussed between reviewers and authors, and between reviewers. After the discussion phases, the scores were 3x borderline reject and 1x weak accept.

The overall consensus is that the paper is a great read and proposes an interesting angle for obtaining equivariant representations. On the negative side, there were several concerns with respect to the experimental setup and if they support the general claims of the work. Some of them were successfully rebutted by the authors:
- the paper does indeed evaluate on a relevant real world task (CO3D camera pose estimation)
- it also successfully shows in the ablations in Section 5.1 that the proposed losses are necessary to recover the correct transformations
- the claim that the method performs on par with hand-designed methods is supported (if only on limited experiments)

However, there are also concerns left, which have not been adequately resolved:
- It is true that the paper lacks theoretical justifications for the many "intuitive" heuristics that it employs
- It is also true that the paper mostly invents its own baselines and datasets. Only the measurements of equivariance error (rebuttal document) seem to be in the same framework as previous works. Meanwhile, the NFT authors provide a benchmark set (https://github.com/masomatics/NFTPublic/tree/main), which is not used by the authors of this work.

Regarding the last point it has to be said that the presented approach is indeed quite novel and the used datasets and evaluation methods seem to be quite creative to me. In this borderline situation, I tend to recommend acceptance, favoring the explorative nature of this work.